# Positional proteomics reveals differences in N-terminal proteoform stability

Daria Gawron[1,2], Elvis Ndah[1,2,3], Kris Gevaert[1,2] & Petra Van Damme[1,2,*]

## Abstract

To understand the impact of alternative translation initiation on a proteome, we performed a proteome-wide study on protein turnover using positional proteomics and ribosome profiling to distinguish between N-terminal proteoforms of individual genes. By combining pulsed SILAC with N-terminal COFRADIC, we monitored the stability of 1,941 human N-terminal proteoforms, including 147 N-terminal proteoform pairs that originate from alternative translation initiation, alternative splicing or incomplete processing of the initiator methionine. N-terminally truncated proteoforms were less abundant than canonical proteoforms and often displayed altered stabilities, likely attributed to individual protein characteristics, including intrinsic disorder, but independent of N-terminal amino acid identity or truncation length. We discovered that the removal of initiator methionine by methionine aminopeptidases reduced the stability of processed proteoforms, while susceptibility for N-terminal acetylation did not seem to influence protein turnover rates. Taken together, our findings reveal differences in protein stability between N-terminal proteoforms and point to a role for alternative translation initiation and co-translational initiator methionine removal, next to alternative splicing, in the overall regulation of proteome homeostasis.

**Keywords** alternative translation initiation; initiator methionine processing; N-terminal proteoform; protein stability; ribosome profiling
**Subject Categories** Genome-Scale & Integrative Biology; Post-translational Modifications, Proteolysis & Proteomics
**Mol Syst Biol. (2016) 12: 858**

## Introduction

Protein stability has been suggested to reflect the individual properties and biological functions of proteins (Cambridge *et al*, 2011; Boisvert *et al*, 2012). In eukaryotic cells, a wide variety of proteins with roles in cell cycle progression, signal transduction and metabolic regulation are quickly degraded by the proteasome (Hershko &

Ciechanover, 1998), whereas some constituents of abundant protein complexes (such as the ribosome and the spliceosome) are extremely stable (Cambridge *et al*, 2011; Boisvert *et al*, 2012). Short-lived proteins are predicted to have an increased aggregation potential, which often places them central in protein deposition diseases correlated with ageing and reduced proteasome activity (De Baets *et al*, 2011). Estimations of the impact of protein turnover on gene regulation revealed that protein synthesis and degradation rates are comparable for proteins within functional protein classes and among individual components of macromolecular protein complexes (Kristensen *et al*, 2013). As such, protein stability is undoubtedly a key component that regulates the outcome of gene expression and shapes cellular phenotypes. Large-scale studies of proteome dynamics in living cells were made possible by applying stable isotope labelling, including SILAC (stable isotope labelling by amino acids in cell culture), in combination with mass spectrometry. To measure protein turnover, cells are usually pulsed with media containing stable isotopic variants of essential amino acids and the rate of incorporation of these newly added isotopes then enables calculating the protein half-life (Doherty *et al*, 2009; Cambridge *et al*, 2011; Schwanhausser *et al*, 2011).

In several recent N-terminomics analyses using mouse and human cell lines, we showed that the prevalence of translation start sites not annotated in protein sequence databases was underestimated as we found that 10–20% of all protein N-termini resulted from alternative translation initiation events, point to inadequately annotated protein start sites or to alternative splicing (Menschaert *et al*, 2013; Koch *et al*, 2014; Van Damme *et al*, 2014). Scarce evidence indicates that alternative translation initiation can produce protein variants—proteoforms (Smith & Kelleher, 2013)—that differ in stability when compared to their database-annotated counterparts (Song *et al*, 2009). As an example, alternative translation initiation of the opioid receptor that results in an N-terminal protein extension rich in lysine residues and prone to ubiquitination contributes to the high degradation rate of this proteoform (Song *et al*, 2009).

Triggered by such atomistic studies, we set out to study on a more global scale if N-terminal proteoforms derived from the same gene showed different stabilities. In contrast to other proteome studies that do not distinguish between N-terminal proteoforms, we here combine SILAC pulse labelling (Boisvert *et al*, 2012) (pSILAC) with

1 Department of Medical Protein Research, VIB, Ghent, Belgium
2 Department of Biochemistry, Ghent University, Ghent, Belgium
3 Lab of Bioinformatics and Computational Genomics, Department of Mathematical Modelling, Statistics and Bioinformatics, Faculty of Bioscience Engineering, Ghent University, Ghent, Belgium
*Corresponding author. Tel: +32 92649279; Fax: +32 92649496; E-mail: petra.vandamme@vib-ugent.be

N-terminal COFRADIC (COmbined FRActional DIagonal Chromatography) (Gevaert *et al*, 2003) to study the stability and turnover rates of proteoforms with heterogeneous N-termini. Important here is that we rely on known (previously identified, Van Damme *et al*, 2014) N-terminal modifications to unambiguously assign an identified peptide as a proxy for a protein's N-terminus. Indeed, N-termini of eukaryotic proteins typically undergo co-translational modifications, which can be used to discriminate N-termini pointing to translation initiation events from those pointing to proteolytic cleavage using mass spectrometry (Van Damme *et al*, 2009). For example, methionine aminopeptidases typically remove the initiator Met (iMet) if the side chain of the next amino acid has a small gyration radius as found in alanine (Ala), valine (Val), serine (Ser), threonine (Thr), cysteine (Cys), glycine (Gly) or proline (Pro) (Frottin *et al*, 2006). Subsequently, the alpha-amino group of the resulting N-terminus, with the exception of Pro, are frequently modified by N-terminal acetylation (Nt-acetylation) catalysed by N-terminal acetyltransferases (Nt-acetyltransferases or NATs) (Arnesen *et al*, 2009). iMet-retaining N-termini may also be Nt-acetylated, largely depending on the sequence specificity of the NAT involved (Van Damme *et al*, 2011, 2012). Taken together, up to 90% of all human proteins undergo either partial or complete Nt-acetylation (Starheim *et al*, 2012). For several cases, Nt-acetylation was shown to alter protein interactions, functions and structure (Skoumpla *et al*, 2007; Scott *et al*, 2011; Shemorry *et al*, 2013; Dikiy & Eliezer, 2014). Moreover, protein N-termini have other unique properties, such as increased disorder in their secondary structure (Holmes *et al*, 2014). Surprisingly, the length of N-terminal disorder strongly correlates with the probability of protein Nt-acetylation, with highly disordered sequences being more frequently Nt-acetylated. As a consequence, it was suggested that Nt-acetylation may play a role in structural stabilisation. More specifically, Nt-acetylation of alpha-synuclein was shown to stabilise the helicity of its N-terminus (Dikiy & Eliezer, 2014) and loss of Nt-acetylation seems to inhibit aggregation of prion proteins (Holmes *et al*, 2014). Moreover, Nt-acetylation may have important roles in protein stability. Although it was long believed that Nt-acetylation may prevent protein degradation by the ubiquitin system (Hershko *et al*, 1984), other studies indicated that Nt-acetylated proteins are not significantly more stable or shielded from proteolytic degradation (Brown, 1979). The N-end rule further explores the relationship between the *in vivo* protein half-life and the identity of a protein's N-terminal amino acid(s). In summary, the N-end rule links the removal of protein substrates covering the whole spectrum of N-terminal identities, except of Pro- and Gly-starting N-termini. Degradation signals encoded by N-terminal residues, referred to as N-degrons, may be recognised by specialised E3 ubiquitin ligases and the N-end rule consists of two major branches, being the Arg/N-end rule and the Ac/N-end rule pathways. The Arg/N-end rule targets unmodified iMet-starting N-termini followed by a hydrophobic amino acid (Kim *et al*, 2014) and neo-N-termini generated upon proteolytic cleavage (Piatkov *et al*, 2012; Kim *et al*, 2014). The acetylation-dependent branch of the N-end rule (Ac/N-end rule) mediates a targeted degradation of proteins with Nt-acetylated Met, Arg, Val, Ser, Thr and Cys residues (Hwang *et al*, 2010). Interestingly, Nt-acetylation of two N-terminal proteoforms raised by alternative initiation of SNC1 in *Arabidopsis thaliana* had an opposing effect on proteoform stability dependent on the NAT involved (Xu *et al*, 2015). The longest

form was destabilised by NatA, whereas NatB-mediated Nt-acetylation increased the stability of the truncated proteoform. On the other hand, recent studies of human Ogden syndrome caused by a NatA loss-of-function mutation indicated that Nt-acetylation was involved in the increased stability of the NatA substrate THOC7 (Myklebust *et al*, 2015). Next to the N-end rule, additional mechanisms have been shown to repress undesirable protein degradation. For example, N-degrons may be shielded by proper folding of the mature protein or by integration into macromolecular complexes (Eisele & Wolf, 2008; Shemorry *et al*, 2013; Park *et al*, 2015).

To summarise, alternative translation initiation and co-translational modifications seem to contribute substantially to the diversity of protein N-termini and are implicated in the regulation of protein turnover. Evidently, a comprehensive characterisation of N-terminal proteoforms linked to their stability is of great importance.

# Results

We combined N-terminal COFRADIC, a positional proteomics technology that enables the enrichment of N-terminal peptides, with pSILAC for the identification of N-terminal proteoforms and characterisation of their cellular stability in human Jurkat T-lymphocytes. Following database searching, we retained N-termini starting at position 1 or 2 that point to "database-annotated" translation initiation sites (dbTIS). Next, we inspected N-terminal peptides starting beyond position 2 and selected those peptides that comply with the rules of iMet processing and Nt-acetylation (Arnesen *et al*, 2009; Van Damme *et al*, 2011). Proteoforms linked to these N-terminal peptides were expected to be mainly derived from the use of alternative translation initiation sites (aTIS).

Overall, we identified 2,578 N-terminal peptides (Table EV1A), including 2,135 dbTIS and 443 aTIS. Our study demonstrates that N-terminal proteoforms raised upon translation initiation and alternative iMet processing contribute substantially to the diversity of the human proteome. Indeed, we identified 335 N-terminal proteoforms derived from 133 genes displaying multiple N-termini raised from different translation initiation sites (Fig 1A). Additionally, we observed 81 proteoforms resulting from partial iMet processing (Fig 1A). Of note, 63% of all aTIS were identified in the absence of their canonical counterparts.

## Diversity of N-terminal proteoforms

Deviations from the canonical translation mechanism such as ribosome leaky scanning and re-initiation (Sonenberg & Hinnebusch, 2009; Hinnebusch, 2011) allow for selection of alternative translation initiation sites within one transcript and, together with mRNA splicing, contribute to the N-terminal proteoform diversity observed in our study. Leaky scanning is a mechanism by which a fraction of ribosomes omits less favourable start codons to initiate translation downstream (Kozak, 2002; Gawron *et al*, 2014). Ribosome profiling previously demonstrated (Lee *et al*, 2012) that leaky scanning is responsible for most database non-annotated TIS selection events and that the efficiency of downstream initiation is heavily dependent on the strength of the translation initiation context surrounding dbTIS, a finding corroborated by our previously

**Figure 1. The origin of N-terminal proteoforms.**

A    2,578 proteoforms were identified including 2,135 dbTIS and 443 aTIS. For the majority of proteins, we found a unique translation initiation site (grey). Additionally, 335 N-terminal peptides pointed to proteins with multiple N-termini created by alternative translation initiation (blue), whereas 81 N-termini were generated by alternative iMet processing (green).

B, C    Proteomics-derived translation initiation events were verified at the transcript level using Ribo-Seq data obtained in Jurkat cells and HCT116 cells (data unpublished). Although in this study, the degree of *in vivo* Nt-acetylation could not be directly determined, all retained N-termini were compliant with the rules of N-terminal processing (Van Damme *et al*, 2014). In the case N-termini were found to be *in vivo* Nt-acetylated (previous MS/MS-based evidence, Van Damme *et al*, 2014), this information is indicated. dbTIS and aTIS peptides were further mapped to protein N-termini annotated in Swiss-Prot Isoform, UniProt TrEMBL and Ensembl databases and neo-N-termini (proteolytic products) of the knowledgebase TopFind 3.0. The number of matching N-termini is indicated in dark grey. N-terminal peptides supported by at least one source of metadata pointing to translation (Nt-acetylation, Ribo-Seq, Swiss-Prot Isoforms, TrEMBL or Ensembl) were classified as highly confident TIS (indicated in green).

D    Using experimental and metadata, we assigned the most likely origin of aTIS peptides and confirmed that alternative translation initiation within the same transcript (leaky scanning) contributes to the majority of alternative TIS identified.

obtained N-terminomics data (Van Damme *et al*, 2014). To study the contribution of (alternative) translation initiation and splicing to N-terminal proteoform expression in more detail, we performed ribosome profiling experiments in Jurkat cells. Additionally, the N-terminal peptides identified in our proteomics experiment in the Swiss-Prot database were mapped to alternative databases: human Swiss-Prot Isoform, UniProt TrEMBL and Ensembl.

Ribosome profiling, or Ribo-Seq, enables studying *in vivo* protein synthesis by deep sequencing of ribosome-associated mRNA fragments, thereby providing a genome-wide snapshot of actively translated mRNAs. Additionally, (alternative) translation initiation can be studies with sub-codon to single-nucleotide resolution through the use of antibiotics such lactimidomycin (LTM), which exclusively inhibit initiating ribosomes (Lee *et al*, 2012).

Using ribosome profiling, of the 2,578 N-terminal proteoforms identified by means of N-terminomics, we were able to confirm translation initiation for 89% of dbTIS (1,895 of 2,135) and 29% of aTIS (130 of 443 aTIS) in Jurkat cells (Fig 1B and C). Using a Mann–Whitney one-sided test, we further confirmed that aTIS detected by Ribo-Seq (130) had significantly lower translation initiation signal ($R_{LTM-CHX}$) at the start codon compared to dbTIS (1,895) (*P*-value = 0.011). The same was observed when directly comparing dbTIS to aTIS variants (31 pairs) detected by Ribo-Seq and originating from the same gene (Wilcoxon signed-rank test for paired samples; *P*-value of 0.00068). Further, from the data mapping to Ensembl and Swiss-Prot Isoforms databases, we found that 23% of the aTIS proteoforms (101 out of 443) could likely be explained by a database-annotated alternative splicing event (Fig 1D). Additionally,

only 19 aTIS N-termini were annotated in the proteolytic knowl-edgebase of protein N-termini, TopFIND 3.0, as potential protease cleavage products (Fortelny *et al*, 2015) (Fig 1C). However, of these seven could be disregarded due to Nt-acetylation evidence and/or Ribo-Seq confirmed TIS selection. Overall, given the N-terminal selection strategies applied and the metadata at hand, only very few of the here reported N-termini can be considered as potential prote-olytic products. The majority of remaining N-terminal proteoforms (330) is thus likely generated upon leaky scanning or other transla-tion initiation mechanism as previously shown by others and us (Lee *et al*, 2012; Van Damme *et al*, 2014). Leaky scanning frequently results in initiation at a downstream start codon in the immediate proximity of the first AUG codon (Fig 2A) (Van Damme *et al*, 2014); however, supported by Ribo-Seq data, we found exam-ples of translation initiation by means of leaky scanning at more distant start codons (Fig 2B). Relying on the current Ensembl human transcriptome annotation and splice variants present in the Swiss-Prot Isoform database, we were additionally able to confirm the expression of N-terminal proteoforms from alternatively spliced transcripts (Fig 2C).

Interestingly, we found 12 aTIS proteoforms being exclusively expressed in the absence of dbTIS, judging from the lack of preced-ing Ribo-Seq and proteomics evidence (Van Damme *et al*, 2014) (Fig 2D). The two most likely scenarios leading to the creation of N-terminal proteoforms starting at two consecutive Met residues (e.g. at positions 1 and 2) are leaky scanning or alternative iMet processing. Our proteomics data provide evidence of consecutive Met-starting N-termini for 11 genes and matching Ribo-Seq data indicate that translation initiation may occur at both Met residues (Fig 2E and F).

**A**  G3BP1 gene: Met at position 1 and 3 are TIS, alternative translation initiation in the same exon

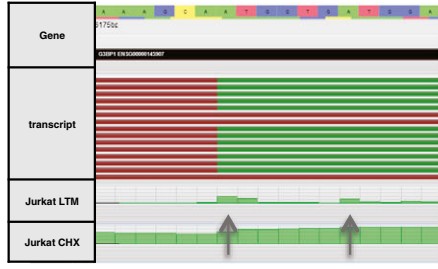

**D**  ACIN1 gene: Met at position 59 is exclusively expressed

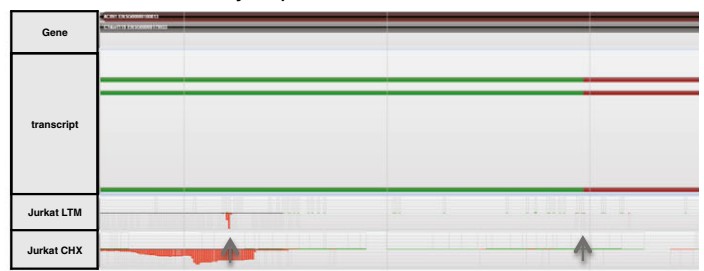

**B**  UBA1 gene: Met at position 1 and 41 are TIS, alternative translation initiation at exon 2 and 3

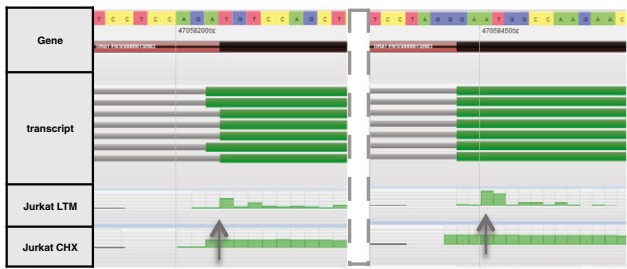

**E**  HNRNPF gene: Met at position 1 and 2 are TIS

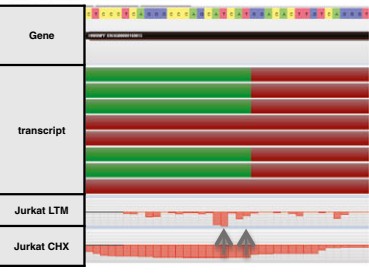

**C**  MAPRE2 gene: Met at position 1 and 44 are TIS, alternative splicing of exon 1 and 2

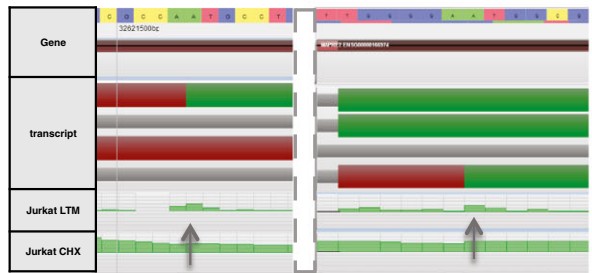

**F**  NUDT4 gene: Met at position 1 and 2 are TIS

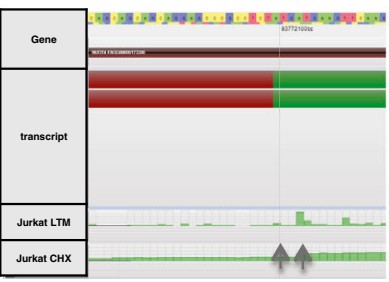

**Figure 2. Examples of translation initiation events confirmed by Ribo-Seq in Jurkat cells.**

A, B   Alternative translation initiation due to leaky scanning.
C        Alternative splicing.
D        The preferential expression of an N-terminally truncated proteoform.
E, F    Evidence for alternative translation initiation occurring at two consecutive AUG codons.

   

**Protein turnover measurement**

To measure turnover times of N-terminal proteoforms, we used human Jurkat T-lymphocytes that were pre-labelled with either light (Arg[0]) or medium (Arg[6]) isotopes of arginine and harvested 0.5, 1.5, 4, 8, 12, 24 or 48 h after a label swap to light or heavy arginine (Arg[10]) containing medium (see Fig 3A). Note that arginine was used as the SILAC label given that by nature of the N-terminal COFRADIC procedure, all N-terminal peptides end on arginine. After mixing equal amounts of light and medium/heavy labelled proteome samples taken at the seven different time points, N-terminal COFRADIC analyses were performed.

From the total of 2,578 N-terminal peptides identified (Table EV1A), we retained 1,972 peptides that were identified in at least three of the time points analysed (Table EV1B). Their SILAC ratio values were then used to calculate protein stabilities by monitoring protein degradation and synthesis based on changes in M/L and H/L isotope ratios, with the crossing point between the degradation and synthesis profiles enabling a direct read-out of the 50% protein turnover (Fig 3B) (Boisvert *et al*, 2012). Using a simple exponential model, we calculated turnover times for 1,928 (98%) of the 1,972 proteoforms with minimal $R^2$ coefficients ≥ 0.8. To evaluate the relationship between the number of observations and the predictive power of our model, we re-calculated turnover times of

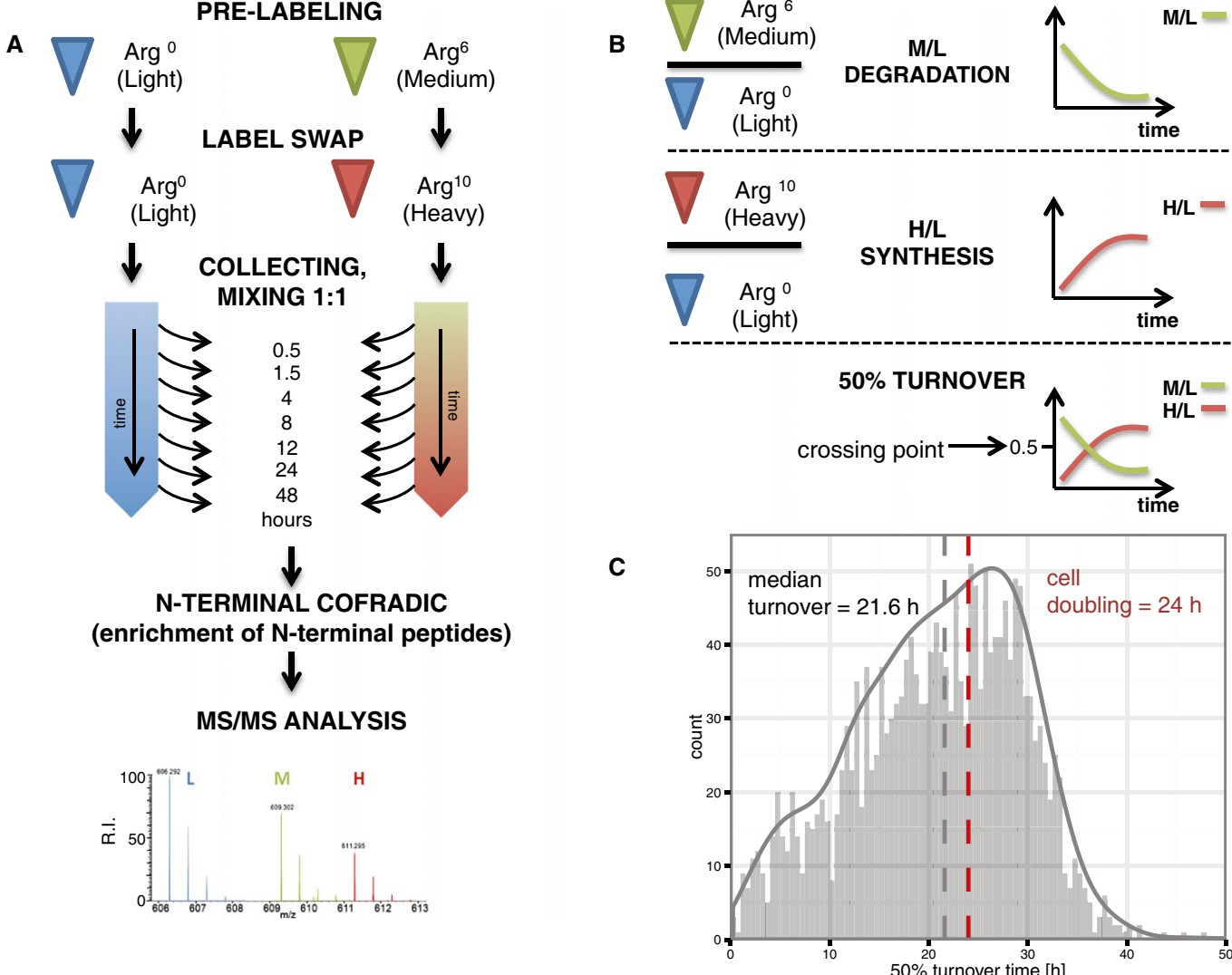

**Figure 3.  Experimental set-up using pSILAC and N-terminal COFRADIC to assess N-terminal proteoform stability.**

A   Jurkat cells were pre-labelled with light or medium L-Arg isotopes. A label swap was performed of cells growing in medium Arg, whereas light cells were cultured in the same medium. Cells were harvested at different time points and equal proteome amounts were mixed, followed by N-terminal COFRADIC and LC-MS/MS analysis.
B   Ratios between three dynamic forms of N-terminal peptides reflect protein synthesis and degradation rates, allowing the calculation of 50% turnover times.
C   The distribution of turnover times (N = 1,894) calculated was not unimodal, with a sizable number of quickly turned over proteins and the median protein turnover rate being 21.6 h. The doubling time of Jurkat cells (24 h) is indicated in red.

selected proteoforms based on a subset of available observations, showing that turnover times can be precisely reproduced using as few as three randomly selected data points (Fig EV1). Our model very well described the variation in experimental data (as supported by a high median $R^2$ value of 0.98; see Fig EV2), overall demonstrating that protein degradation in Jurkat cells follows mono-exponential kinetics. Only a minority of degradation and synthesis curves (2%) showed monotonic changes in time not following an exponential trend ($R^2 < 0.8$). In 13 such cases, turnover times were determined using a linear interpolation of the crossing point, but otherwise such proteoforms were excluded from further analysis. In total, we could assign 50% turnover times to 1,941 N-terminal proteoforms (Table EV1B).

To account for cell growth, our exponential model of protein degradation incorporated the doubling time measured for Jurkat cells (24 h, see Fig EV3). This allowed for the determination of two additional parameters, namely protein degradation rate constants ($k_{deg}$) and protein half-life values [h]. Since 50% turnover values can be measured more directly and reflect both protein synthesis and degradation, we selected 50% turnover time as the desired parameter to perform our subsequent analyses.

**Protein function and turnover: correlation with previous reports**

Overall, protein turnover times in Jurkat cells varied from < 1, to more than 48 h with a median turnover time of 21.6 h (Fig 3C). Protein degradation constants determined for Jurkat cells (control sample) (Fierro-Monti *et al*, 2013), protein turnover times measured in HeLa cells (whole cell lysates) (Boisvert *et al*, 2012) and protein half-lives derived from NIH3T3 mouse fibroblasts (Schwanhausser *et al*, 2011) were retrieved and compared to their corresponding values of canonical proteoforms quantified in our study. We detected a moderate correlation between our data and previous reports, with Spearman rank correlation coefficients reaching 0.49 for HeLa data (Boisvert *et al*, 2012), 0.48 for NIH3T3 data (Schwanhausser *et al*, 2011) and 0.44 for Jurkat data (Fierro-Monti *et al*, 2013). Moreover, a median protein half-life of 50.5 h measured in our study was in good agreement with previously reported values in mouse NIH3T3 cells (47.8 h) (Schwanhausser *et al*, 2011) and human Jurkat cells (55.8 h) (Fierro-Monti *et al*, 2013). Further, we observed a highly similar median turnover time (21.6 h) and turnover distribution to estimates obtained in HeLa cells (20 h) (Boisvert *et al*, 2012).

Based on gene ontology (GO) biological process term enrichment of our data using the Gorilla software (Eden *et al*, 2009) and considering one proteoform per gene, we found that proteins with common functional annotations displayed comparable cellular stabilities. To reduce the complexity and redundancy of enriched categories, we applied REViGIO (Supek *et al*, 2011) and visualised selected GO terms with false discovery rate (FDR) q-values lower than 0.05. Unstable proteins were more often involved in mitosis, chromosome segregation, cell cycle and apoptosis, next to regulation of RNA transcription and protein ubiquitination (Fig 4A). On the other hand, RNA splicing, protein translation, folding and transport as well as various metabolic processes of carbohydrates, nucleotides, aldehydes, ketones, glutathione, nitrogen and phosphorus were conducted by more stable proteins (Fig 4B). Along the same line, category enrichment analysis of KEGG pathway terms clearly pointed to increased stability of spliceosome and ribosome components next to RNA degradation, aminoacyl-tRNA biosynthesis, glycolysis and gluconeogenesis, and pentose phosphate pathways (all with FDR values below 9.0E-04) (Schwanhausser *et al*, 2011; Boisvert *et al*, 2012). Pfam data did not point to a significant influence of protein domain composition on protein turnover.

Overall, the results of our annotation enrichment analysis agree very well with previous studies in mouse (Schwanhausser *et al*, 2011) and human (Boisvert *et al*, 2012; Fierro-Monti *et al*, 2013), and thus, protein turnover rates seem to be conserved between different cell lines and organisms, especially at the level of protein groups and biological processes.

**Stability, abundance and structural disorder of alternative proteoforms**

To facilitate the interpretation of the relationship between alternative translation and protein turnover, whenever an iMet-retaining and iMet-processed proteoform were identified, we only selected the expected mature N-terminus per proteoform, according to the rules of iMet processing (Frottin *et al*, 2006). In total, we assigned protein turnover rates for 323 aTIS and 1,571 dbTIS N-terminal proteoforms (Fig 5A).

The majority of the identified aTIS resulted in relatively short N-terminal protein truncations (protein length reduced by less than 13%). The distribution of aTIS corresponding iMet positions centred within the first 200 amino acids with a median position of 52 (Fig 5B), an observation in line with the fact that leaky ribosome scanning is likely to be responsible for the majority of downstream translation events observed (see above and Lee *et al*, 2012; Van Damme *et al*, 2014).

Interestingly, N-terminal proteoforms derived from aTIS usage frequently displayed distinct stabilities (Fig 5C). We were able to quantify turnover rates of 100 dbTIS/aTIS pairs and observed that some aTIS displayed altered stabilities as compared to their dbTIS counterparts. More specifically, we detected numerous examples of aTIS proteoforms with increased as well as decreased stability (Fig 6A and B). This effect appeared not to correlate with the distance between aTIS and dbTIS positions (Fig 6C). Since the differences in stabilities between aTIS and dbTIS expressed forms of the same gene were not uniform, no statistical difference was detected between their turnover times (Fig 6C, Table 1). However, when we compared the entire populations of aTIS to dbTIS, alternative proteoforms were on average found to be significantly more stable (*P*-value = 1.52e-08) (Fig 5C, Table 1).

We further determined whether aTIS indicative N-termini share features that are distinct from dbTIS. Given that translation is a one-directional process and downstream initiation due to leaky scanning only occurs with a limited probability, in a normal cellular context one might expect that many aTIS variants are less efficiently produced than dbTIS variants, leading to lower abundance of the former (Michel *et al*, 2014; Van Damme *et al*, 2014). To investigate this, we used spectral counts as well as NSAF scores (Paoletti *et al*, 2006) as quantitative proteoform measures. Using a Mann–Whitney one-sided test to compare the aTIS (323) and dbTIS (1,571) groups, we confirmed that, on average, aTIS variants are less abundant with high significance values when using both spectral counts (*P*-value = 4.31e-09) and NSAF scores (*P*-value = 1.15e-07). The

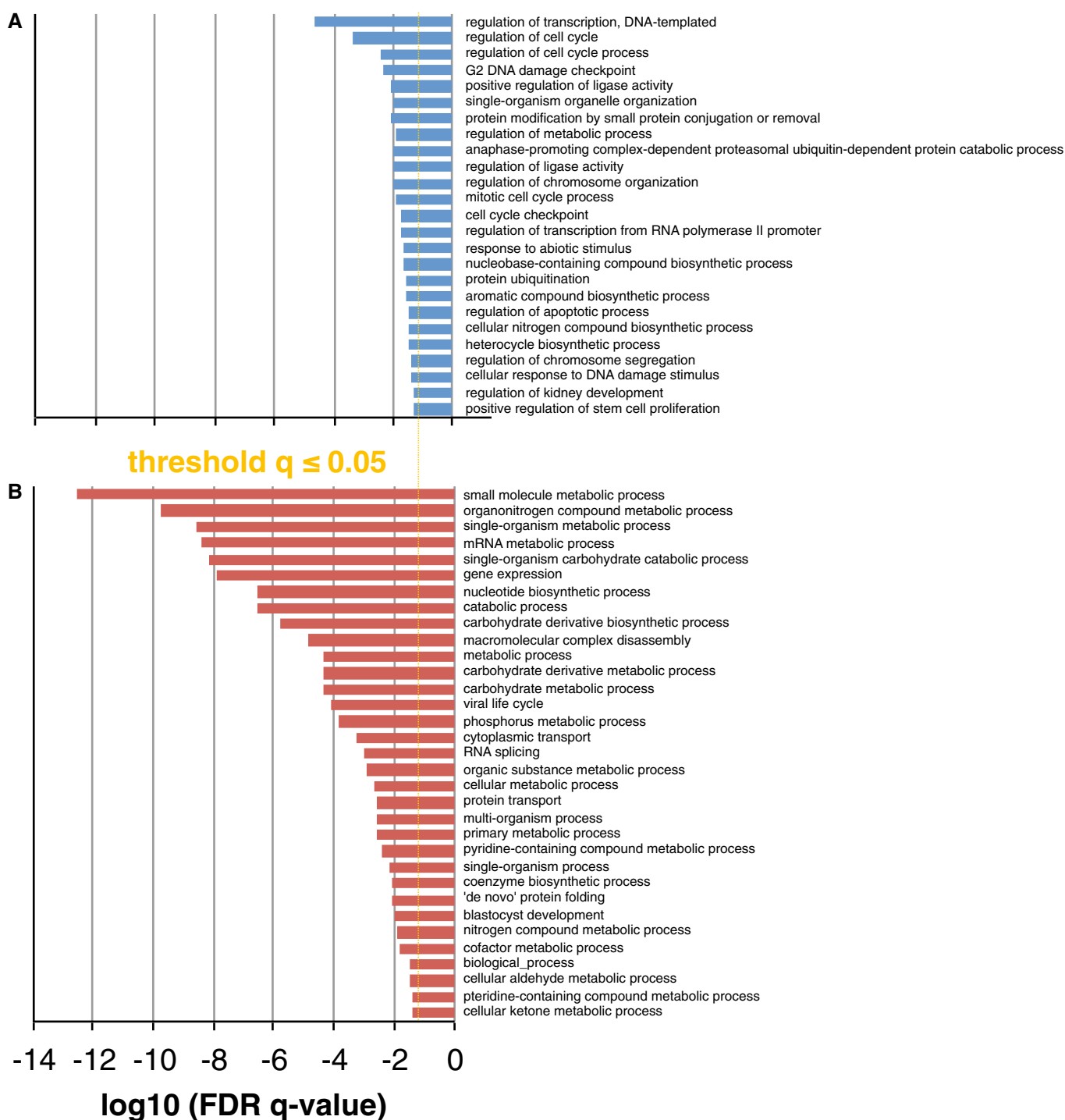

**Figure 4.    GO term enrichment analysis for unstable (A) and stable (B) proteins.**

A, B   Horizontal bar chart representations are given for significantly enriched GO terms in the human proteome (FDR *q*-value ≤ 0.05).

same was observed when directly comparing dbTIS to aTIS variants originating from the same gene using a Wilcoxon signed-rank test for paired samples (*P*-values of 3.44e-09 and 6.45e-09, respectively, for spectral counts and NSAF, see Table 1). Finally, we observed a slight positive correlation between turnover time and protein abundance (Spearman rank correlation of 0.285). Further analysis indicated reduced levels of the top 10% of unstable proteins and higher levels of the 10% most stable proteins in relation to the mean protein abundance in our data set (Benjamini–Hochberg FDR of 2.29e-11 and 7.44e-05, respectively).

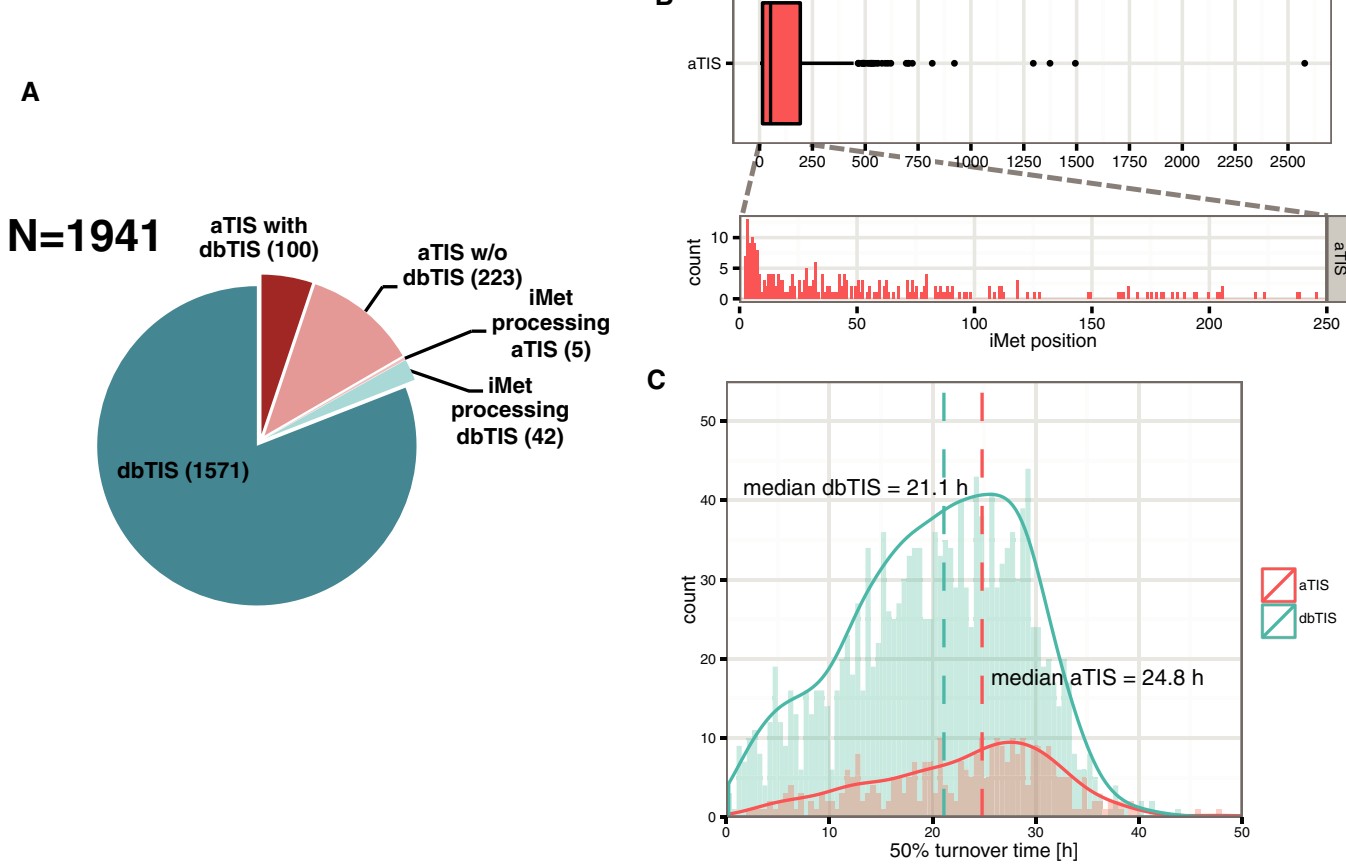

**Figure 5.   Characterisation of identified N-terminal proteoforms.**

A   We assigned the turnover time of 1,941 proteoforms, including 1,894 proteoforms derived from 1,571 dbTIS and 323 aTIS next to 47 N-termini generated by alternative iMet processing. dbTIS- and aTIS-derived proteoforms are indicated in shades of blue and shades of red, respectively. Stability of 100 N-terminal proteoform pairs derived from the same gene (dark red).

B   Distribution of iMet positions giving rise to 323 N-terminal proteoforms with assigned turnover times (to avoid redundancy a single iMet-processed proteoform was considered).

C   Distribution of turnover values shows differences in the overall stability of dbTIS versus aTIS proteoforms.

Intrinsic disorder in protein structures is an important factor that determines protein degradation rates (reviewed in Babu *et al*, 2011). The extent of structural flexibility can be assessed, among others, by studying the amino acid composition of proteins (Dosztanyi *et al*, 2005b). For every proteoform sequence, the percentage of intrinsically disordered regions was predicted using the RAPID server (Yan *et al*, 2013). The average disorder of aTIS proteoforms was found to be higher than that of dbTIS proteoforms (Mann–Whitney one-sided test, $P = 0.00157$). These observations were confirmed in paired samples of aTIS and dbTIS products of the same gene (Wilcoxon signed-rank test, $P = 0.01216$, see Fig 6D). In conclusion, aTIS proteoforms were characterised by lower expression levels and higher structural flexibility than dbTIS proteoforms. Moreover, the aTIS population displayed an overall higher stability compared to dbTIS. Additionally, a low negative correlation was observed between protein turnover and the percentage of disordered sequence (Spearman rank correlation of −0.209), suggesting that unstable proteins tend to have more disordered regions, as reported in previous studies (Doherty *et al*, 2009; Kristensen *et al*, 2013).

In line with this, the 10 % most stable proteins had significantly lower disorder content, contrary to the 10 % least stable proteins with highly disordered structures (Benjamini–Hochberg FDR of 3.27e-11).

**Protein turnover and N-terminal amino acid composition**

We next investigated whether the identified N-termini hold intrinsic information on the so-called N-degrons (Kim *et al*, 2014), and what the role is of iMet cleavage in this process.

To correlate the N-end rule with our pSILAC data set, we examined the relation between the turnover rate and the identity of the two utmost N-terminal residues. A Bartlett test indicated insufficient homogeneity of variance between the studied groups, leading to the selection of non-parametrical tests for further analysis. First, we performed a pairwise comparison of protein stability across N-termini grouped according to their N-terminal amino acid identity. A Kruskal–Wallis rank sum test indicated a significant impact of N-terminal residues on protein turnover, whereas a subsequent

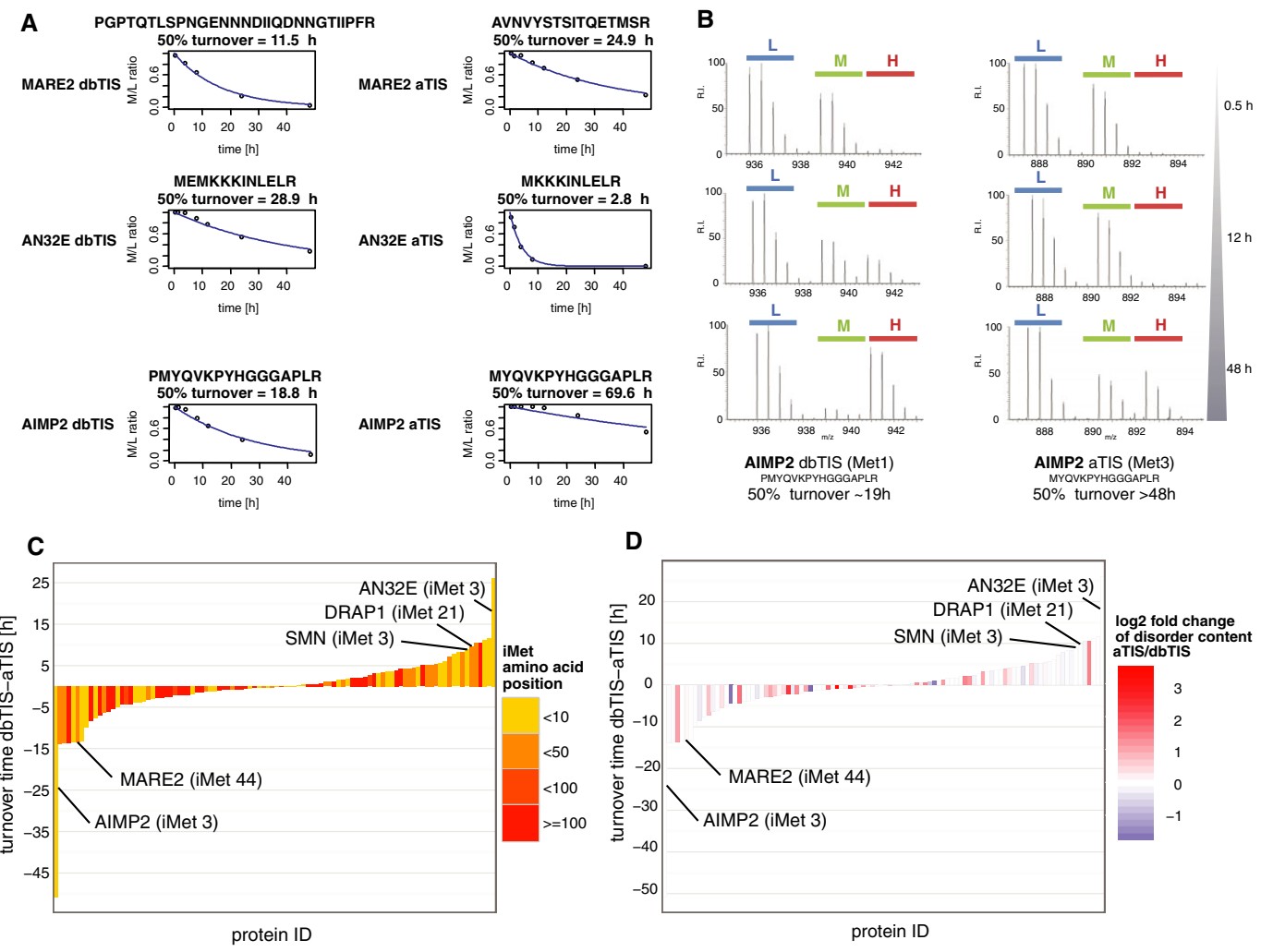

**Figure 6.  N-terminal proteoforms may display different stabilities.**

A   Degradation profiles of AIMP2, MARE2 and AN32E dbTIS and aTIS proteoforms as representative examples of N-terminal proteoforms that differ in stability.

B   MS spectra of the N-termini of two proteoforms originating from the aminoacyl-tRNA synthase complex-interacting multifunctional protein 2 (AIMP2): a dbTIS indicative N-terminus with processed iMet and starting at the second amino acid Pro, an aTIS indicative peptide starting at position 3 and retaining its iMet. As evident from the MS spectra shown, these N-terminal proteoforms displayed largely different turnover rates.

C   Differences in turnover times were calculated for 100 dbTIS and aTIS proteoform pairs and presented in ascending order. The extent of N-terminal truncation in each pair was colour-coded to visualise the absence of correlation between the number of lacking amino acids and the direction of turnover shift.

D   An increase in disorder content between aTIS and dbTIS proteoforms can be observed for a multitude of dbTIS/aTIS pairs.

multiple comparison test allowed to indicate the most deviating pairs (with *P* < 0.05). Met-Asn and Met-Asp were among the least stable N-termini which deviated significantly from stable N-termini such as Met-Thr-, Met-Val-, Met-Ala- and Pro-starting N-termini (Fig 7A).

To further dissect our data, we compared the stability of N-termini undergoing iMet processing, leading to N-terminal Ser, Ala, Thr, Cys, Val, Gly and Pro. Interestingly, a general lower stability was observed for proteoforms with their iMet residue removed compared to proteoforms that retained the iMet. A significant impact of iMet processing on turnover rates was confirmed using a Kruskal–Wallis rank sum test (*P* < 2.2e-16). To decouple the effect of Nt-acetylation from iMet processing, we compared completely or partially Nt-acetylated (Ser, Ala, Thr, Cys, Val, Gly) to *in vivo* free

(Pro) N-termini, and for both groups, we observed a lower stability of proteoforms with their iMet removed (see Fig 7B). Considering individual amino acids, only Thr and Val deviated significantly from their corresponding iMet-retaining N-termini (see Fig 7B and C). However, a less strict analysis with pairwise Wilcoxon rank sum test (without correction for multiple testing) pointed to significant deviation between iMet-processed and iMet-retaining Ala-, Ser- and Gly-starting N-termini. To conclude, turnover rates of N-termini with different susceptibility for Nt-acetylation (Van Damme *et al*, 2011) were generally affected by iMet processing.

These findings suggest that irrespective of (the degree of) Nt-acetylation, iMet processing, rather than Nt-acetylation, seems to be a main contributing factor to N-terminal proteoform stability. In particular and in line with previous findings (Brown, 1979), no

**Table 1.  Differences in turnover, disorder and abundance values between aTIS and dbTIS.**

| Variable | Data set | Test | *P*-value | Conclusion |
|---|---|---|---|---|
| Turnover time | Complete data set | Mann–Whitney test | 1.52e-08 | aTIS>dbTIS |
| | dbTIS/aTIS pairs | Wilcoxon signed-rank test | 0.5021 | No difference |
| Disorder | Complete data set | Mann–Whitney test | 0.00157 | aTIS>dbTIS |
| | dbTIS/aTIS pairs | Wilcoxon signed-rank test | 0.01216 | aTIS>dbTIS |
| Spectral counts | Complete data set | Mann–Whitney test | 4.31e-09 | aTIS<dbTIS |
| | dbTIS/aTIS pairs | Wilcoxon signed-rank test | 3.44e-09 | aTIS<dbTIS |
| NSAF | Complete data set | Mann–Whitney test | 1.15e-07 | aTIS<dbTIS |
| | dbTIS/aTIS pairs | Wilcoxon signed-rank test | 6.45e-09 | aTIS<dbTIS |

For the complete data set of aTIS and dbTIS with reliable turnover measurements and considering a single iMet proteoform ($N$ = 1,894; 323 aTIS and 1,571 dbTIS), a comparison of turnover, disorder and abundance values was performed using Mann–Whitney one-sided test for independent samples. Differences in paired dbTIS-aTIS data were measured using a Wilcoxon signed-rank one-sided test for paired samples. All analyses were performed at $P \leq 0.05$ significance level.

correlation was found with the Nt-acetylation susceptibility of protein N-termini (Fig 7B). Secondly, the presence of a hydrophobic residue at the second position has no apparent influence on average turnover rates as proteins starting with N-termini such as Met-Leu, Met-Phe, Met-Tyr and Met-Ile have stabilities close to the median stability of all protein N-termini (Fig 7A). A more detailed analysis of grand average of hydropathy (GRAVY) of 2 and 10 most N-terminal residues revealed no correlation with protein turnover times (Spearman correlation coefficients of 0.013 and 0.0085, respectively).

**Ubiquitination and protein turnover**

Post-translational modifications, such as ubiquitination, may affect protein stability and are thus worthwhile to consider in the context of N-terminal proteoforms. Ubiquitination has a known, direct impact on protein degradation and was previously studied at the proteome-wide level using Jurkat cells as a model system (Stes *et al*, 2014). Our group reported on a compilation of more than 7,500 *in vivo* ubiquitinated peptides in more than 3,300 different proteins (Stes *et al*, 2014). Using this data set, we investigated whether annotated N-terminal proteoforms holding possible (previously identified) ubiquitination sites displayed different stabilities than proteoforms not found to be ubiquitinated. We compared the stability of 348 dbTIS containing at least one detected ubiquitination site to 388 N-termini with one or more lysine residues but no experimental evidence of ubiquitination in Jurkat cells. A one-sided Mann–Whitney *U*-test revealed a significant higher median stability of ubiquitinated proteoforms compared to non-ubiquitinated

proteoforms ($P$ = 0.0005; see Fig EV4), although in each case a very broad range of protein stabilities could be observed. To provide more insight, we decided to investigate particular cases of ubiquitinated N-termini. The existence of additional ubiquitination sites may contribute to the difference in turnover observed between longer and shorter proteoforms, as reported in the case of the mu opioid receptor (MOP) (Song *et al*, 2009). Interestingly, in our data set, we found 31 dbTIS proteoforms holding unique ubiquitination sites not present in a shorter proteoform (aTIS). Eleven such proteoforms displayed reduced stability (turnover time decreased by at least 2 h) compared to their aTIS counterparts (Table EV2). For these proteoforms, ubiquitination potentially contributes to the observed differences in stability. We also found 14 proteoform pairs where ubiquitination of the longer variant possible contributed to an increased stability and five cases were irrelevant differences in stability between proteoform pairs were observed (< 2 h).

**Protein turnover in macromolecular complexes**

Following up on a recent report on the precise regulation of protein synthesis rates being proportional to the stoichiometry of macromolecular protein complexes (Li *et al*, 2014), we here investigated whether protein synthesis and degradation are coordinated for members of known macromolecular protein complexes (Table EV1B). According to the strategy reported by Kristensen *et al* (2013), we retrieved the human core protein complexes from the CORUM database (Ruepp *et al*, 2010). Next, we assigned turnover times to members of complexes using only the dbTIS proteoforms

**Figure 7.  Relationship between N-terminal amino residue(s) identity and 50% turnover rates.**

A  All protein N-termini were grouped according to the identity of their N-terminal residues and sorted in ascending order of stability.

B  Methionine processing typically affects N-termini displaying amino acids with small gyration radii (S, A, T, C, V, G or P residues at the second position). Although the majority of these N-termini undergo methionine cleavage, some (partially) retain their first amino acid. Subsequently, iMet-processed N-termini may be Nt-acetylated by NatA in the following order of efficiency (S, A, T, C, V and G) with the exception of P which always remains unmodified (Van Damme *et al*, 2011). The stability of iMet-processed N-termini is very similar for S-, A-, T-, C- and G- and P-starting N-termini, but their iMet-retaining counterparts have overall higher stability, and this especially for MT and MV N-termini. In contrast to methionine processing, Nt-acetylation susceptibility does not appear to have an effect on the stability of proteoforms. *The extent of *in vivo* Nt-acetylation was described by Van Damme *et al* (2011).

C  Differences in turnover times where calculated for N-termini starting with S, A, T, C and G and P and their iMet-retaining counterparts (47 proteoform pairs with valid turnover measurements). Decreased stability of processed N-termini could be observed, especially for T- and V-starting N-termini.

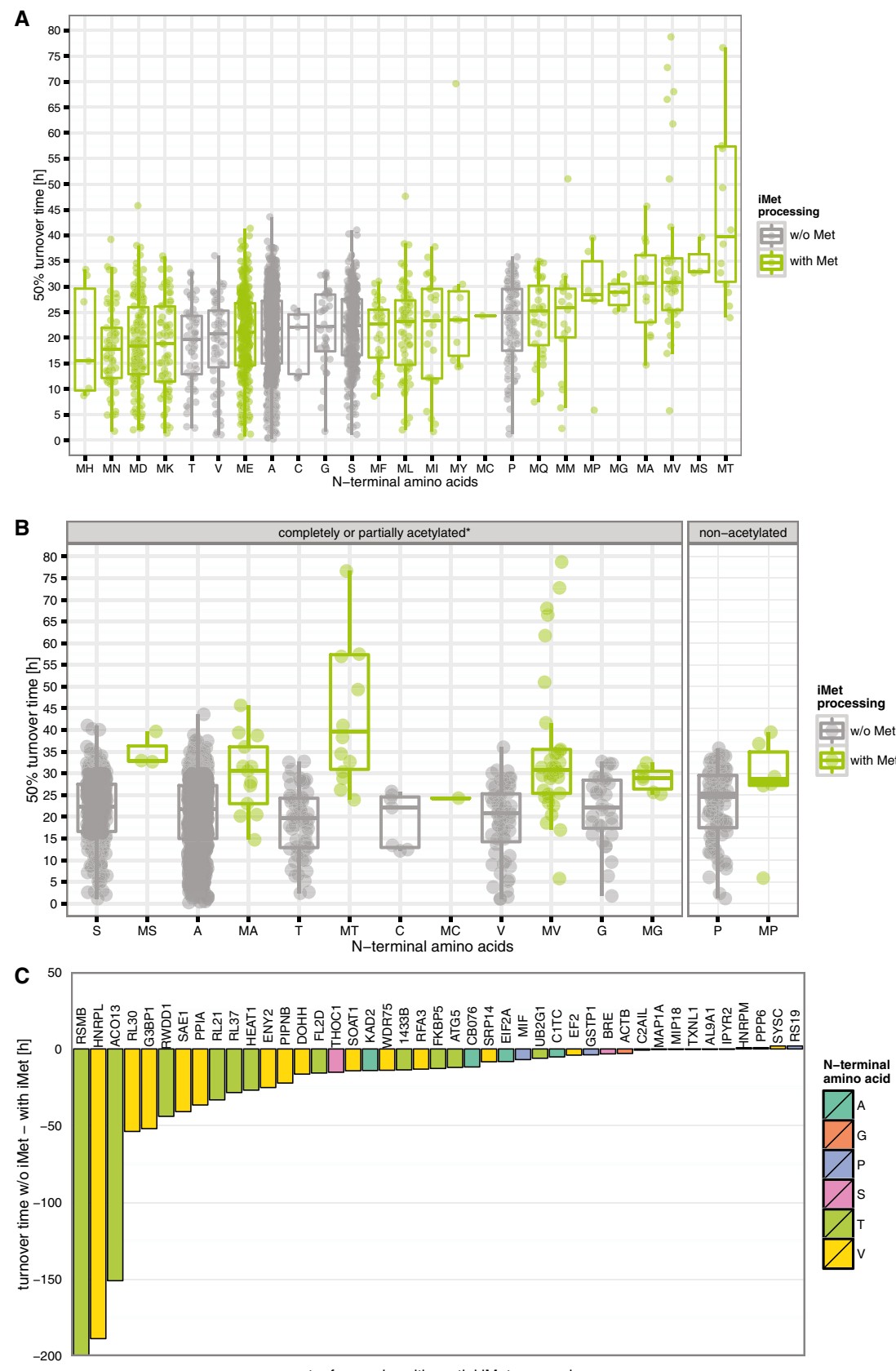

**Figure 7.**

identified in our study. We calculated the absolute difference in stability between every protein and all other members of a given complex. Significant outliers with statistically larger differences to other proteins in a complex were identified using the Mann–Whitney *U*-test (α = 0.05, one-sided). In this way, we estimated the dispersion of turnover rates for 189 proteins within 173 macromolecular complexes with three or more members identified, and came to the conclusion that a majority of turnover rates correlate well within a complex (127 or 73% of the complexes). Inconsistencies were observed in 46 complexes of which 74% involved a single outlier. In 28 complexes, we detected a single unstable component, hinting to a rate-limiting factor for complex formation.

Next, we investigated whether protein–protein interactions could be affected by the expression of alternative N-terminal proteoforms. Here, a comparative analysis of GO terms related to protein function in aTIS- versus dbTIS-derived proteoforms revealed that alternative proteoforms were clearly enriched in protein forming homomultimers (FDR *q*-value of 0.0096) hinting to the likely impact of alternative proteoform incorporation on the overall stability of protein complexes.

**Verification of protein turnover times**

Cycloheximide (CHX) pulse-chase experiments were performed to confirm the stability of selected proteoforms. Half-lives of six selected endogenous proteins (Fig 8A) were assayed by Western blotting using lysates of Jurkat cells treated with CHX for 0–48 h. Protein levels of short-lived securin, lamin B, β-catenin and GCIP-interacting protein p29 levels were monitored by Western blotting and compared to those of stable proteins such as actin and GAPDH. Protein half-lives deduced from Western blot analyses corresponded well to those calculated from pSILAC data. Overexpression of selected V5-tagged proteoforms in the human colorectal HCT116 carcinoma cell line allowed us to validate differential turnover of identified dbTIS and aTIS products from two different genes: MARE2 and AN32E (Fig 8B and C). Data from the latter experiments further suggest that common trends can be observed in the regulation of proteoform stability across different human cell lines, despite discrepancies in the exact values calculated using CHX-chase and pSILAC.

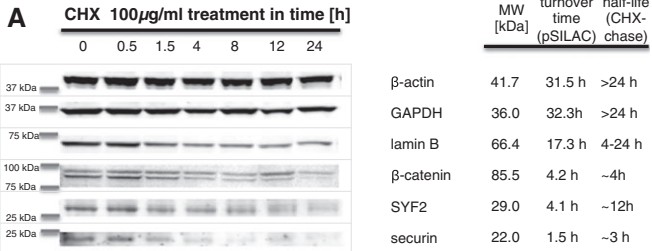

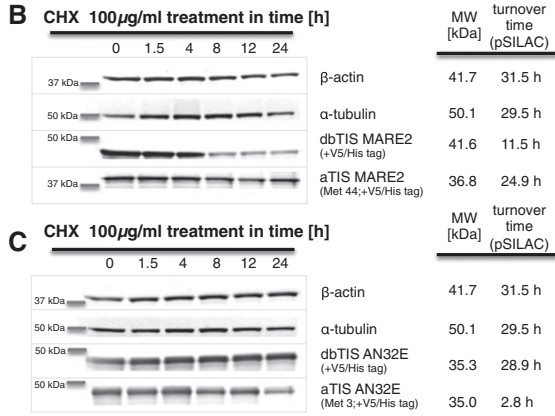

**Figure 8. Comparison of protein turnover measured by pSILAC and CHX-chase.**

A     Jurkat cells were treated with 100 µg/ml CHX for 0, 0.5, 1.5, 4, 8, 12 or 24 h. Protein degradation was monitored by Western blotting, and stabilities of several short-lived endogenous proteins were confirmed using antibodies (including lamin B, securin, β-catenin, GCIP-interacting protein p29) and compared to stable proteins (such as GAPDH and actin). Proteoform-specific bands were used to calculate protein half-lives which were in a good agreement with turnover times obtained from pSILAC

B, C  Validation of the differential turnover time of the dbTIS- and aTIS-derived proteoforms of MARE2 and AN32E. Selective C-terminal V5-tagged proteoforms were overexpressed in HCT116 cells for 24 h, and CHX pulse-chase experiments were performed as described for Jurkat cells. Degradation of overexpressed proteoforms was monitored by an anti-V5 antibody and compared to the turnover of stable proteins such as actin and tubulin.

Source data are available online for this figure.

# Discussion

The selection of translation initiation sites attributes to the complexity of proteomes as N-terminally truncated and extended proteoforms may be differentially regulated, localised (Leissring *et al*, 2004; Kobayashi *et al*, 2009), display altered stabilities (Song *et al*, 2009) and functionalities (Calligaris *et al*, 1995; Thomas *et al*, 2008; Khanna-Gupta *et al*, 2012) especially in the case of some large N-terminal truncations. Proteoforms arisen from alternative translation initiation are generally well conserved among eukaryotic orthologous protein species (Van Damme *et al*, 2014). Selection between alternative translation start sites is regulated under conditions of cellular stress (Gawron *et al*, 2014) and may change the repertoire of N-terminal proteoforms expressed in a developmental and tissue-specific manner (Calligaris *et al*, 1995; Thomas *et al*, 2008).

pSILAC was already used in several proteome studies to determine the dynamic properties of thousands of proteins in a single experiment. Most studies made use of non-synchronised, proliferating cell lines (Selbach *et al*, 2008; Doherty *et al*, 2009; Schwanhausser *et al*, 2011; Boisvert *et al*, 2012), while others employed growth arrest (Cambridge *et al*, 2011) or studied protein synthesis and degradation rates in dynamically changing systems [e.g. during differentiation (Kristensen *et al*, 2013) or following drug-mediated inhibition of Hsp90 (Fierro-Monti *et al*, 2013)]. We decided to employ a pSILAC approach similar to the one described by Boisvert *et al* (2012) which enabled the measurement of protein half-lives ($t_{1/2}$) as well as protein turnover, reflecting the overall protein synthesis and degradation rates (Boisvert *et al*, 2012). To better reflect our experimental results, we simplified the mathematical model proposed by Boisvert *et al* (2012). In their set-up, levels of medium Arg were retained at 20% instead of asymptotically

approaching zero as degradation of proteins proceeds, an observation which they attributed to amino acid recycling. In Jurkat cells, we observed a complete isotope replacement, similar to another recent pSILAC study conducted in the same cell line (Fierro-Monti *et al*, 2013).

Here, we performed a first global, N-terminal proteoform-specific analysis of protein stability and identified numerous N-terminally truncated proteoforms with distinct turnover rates compared to their full-length counterparts. Among the compared dbTIS/aTIS pairs, the stability of alternative proteoforms was not correlated to the length of truncation. The regulation of protein turnover rather seemed proteoform-specific, pointing to many different factors involved in the synthesis and degradation of specific proteins. One such contributing factor may be post-translational modifications, as we found 11 proteoform pairs where ubiquitination of the longer variant possible contributed to reduced proteoform stability.

To validate the difference in half-lives of selected proteoform pairs, we turned to the use of a more classic method for measuring protein turnover, being CHX pulse-chase experiments followed by Western blot analysis. We confirmed turnover rates of several endogenously expressed short- and long-lived proteins. Due to the limitation in molecular weight resolution and sensitivity, we subsequently turned to overexpression studies for the validation of differential N-terminal proteoform stabilities from two human genes. Nonetheless, since upon incubation of the cells with protein synthesis inhibitors, protein degradation may be altered (Hanna *et al*, 2003; Dai *et al*, 2013), and cell toxicity can only be controlled for a relatively short period; CHX treatment may preclude a correct delineation/validation of long protein half-lives.

We further evaluated the origin of N-terminal proteoforms identified in our study using ribosome profiling and extensive mining of public databases (Table EV1C) and found that 74% of the aTIS were generated by alternative translation initiation, whereas 23% likely pointed to splicing events. Truncated proteoforms displayed a lower translation initiation signal at the start codon, in line with their reduced proteome abundance estimated by spectral counting. These finding are consistent with a lower frequency of downstream translation initiation assumed from the leaky scanning model of translation initiation and corroborated by previous reports (Lee *et al*, 2012). Next to their lower abundance, the limited overlap between aTIS identified by means of proteomics and Ribo-Seq might be additionally attributed to the TIS calling approach used, since typically a higher interference is observed at internal TIS typically flanked by signal at both 5′ and 3′ end. Moreover, some aTIS may actually be the sole or preferential TIS site selected due to alternative splicing and, in some cases, alternative translation initiation. In this context, previous studies demonstrated that their corresponding dbTIS sites are typically embedded in less optimal Kozak consensus sequences (Van Damme *et al*, 2014). Indeed, we found 12 aTIS proteoforms exclusively expressed in the absence of dbTIS at both the translatome and the proteome levels. In some cases, a truncated proteoform may also display superior stability over the canonical proteoform. Viewing their general lower abundance, stable aTIS proteoforms are more likely to be detected. The latter could explain why the aTIS population was slightly enriched for long-lived proteins.

As reported previously (Cambridge *et al*, 2011; Kristensen *et al*, 2013), turnover rates were similar for proteins belonging to similar

functional groups or macromolecular complexes. This is supported by the notion that some proteins get stabilised upon complexation with other proteins, lipids, sugars or nucleotides (Cambridge *et al*, 2011). In line, other studies suggested that protein subunits turn over quickly where the complex is assembled, but are stabilised in cellular compartments where the complex is functional (Boisvert *et al*, 2012). Some of the identified protein truncations are predicted to be benign for a protein's function or binding but have large consequences for protein stability. For example, a single amino acid truncation of AIMP2 increases protein stability (see Fig 6A and B) but was not predicted to alter its sequence disorder content, localisation [TargetP (Emanuelsson *et al*, 2000), WoLF PSORT (Horton *et al*, 2007)] or any of its known interacting domains. Considering such cases, a differential regulation of N-terminal proteoform stability may potentially result in elimination or introduction of unstable components into a complex. It is also important to consider that the relative expression levels of proteoforms are responsive to cellular conditions (Touriol *et al*, 2003; Thomas *et al*, 2008; Leprivier *et al*, 2014). In this context, depletion of one proteoform may increase the impact of another proteoform's turnover on protein complex formation. As we observed that aTIS proteoforms were enriched in proteins forming homomultimers, generation of alternative proteoforms may be important in the context of homomultimerisation. Interestingly, in the case of the transcription factor GATA-1, proteoforms generated by alternative translation initiation were already shown to produce dimers (Calligaris *et al*, 1995) with altered activities dependent on the relative ratios of the two proteoforms in the complex.

The extent of intrinsic disorder in protein structures was correlated with turnover times, indicating that a change in structural flexibility of protein N-termini from truncated proteoforms might have an impact on the proteoforms' cellular stabilities. In line with the observation that proteins displaying iMet cleavage are overrepresented *in vivo*, the general belief is that retained iMet reduces the stability of proteins (Meinnel *et al*, 2005; Frottin *et al*, 2006). Unexpectedly, our stability data showed a reduced stability of iMet-processed proteoforms in numerous cases. However, it is important to discriminate between the effect of iMet removal on protein abundance and the effect on protein turnover time, as these two terms should not be used interchangeably. Relying on previous quantitative data on protein Nt-acetylation (Van Damme *et al*, 2014), another common co-translational modification, Nt-acetylation seemed to have no overall impact on turnover rates. Of note however, due to the experimental methodology applied, we cannot distinguish Nt-acetylated from Nt-free proteoform counterparts. In fact, the role of Nt-acetylation in protein stability is debated (Brown, 1979; Hershko *et al*, 1984) and is likely protein context and NAT dependent (Myklebust *et al*, 2015; Xu *et al*, 2015). It was postulated that Nt-acetylation may stabilise otherwise flexible structures of N-termini and reduce their propensity for aggregation (Holmes *et al*, 2014). At the same time, acetylated protein N-termini are substrates of the N-end rule pathway (Hwang *et al*, 2010) which may constitute a clearance mechanism for N-terminally misfolded or non-complexed proteins (Eisele & Wolf, 2008; Heck *et al*, 2010; Shemorry *et al*, 2013; Park *et al*, 2015). Moreover, it is difficult to directly compare our results to the destabilising properties of the corresponding N-terminal amino acids originally described by Varshavsky and colleagues, especially considering differences in the

studied organisms and the techniques used for protein turnover determination (pSILAC vs. CHX-chase). We believe that the property of iMet retention to increase the stability of N-terminal proteoforms described in our study may exist as an additional mechanism next to the N-end rule pathways to ensure the fine-tuning of protein stability in eukaryotic cells.

Recent large-scale studies on protein stability report a limited correlation to other high-throughput data and especially differences in the methodologies applied appear to account for the observed discrepancies (Yen *et al*, 2008; Doherty *et al*, 2009; Cambridge *et al*, 2011; Eden *et al*, 2011; Boisvert *et al*, 2012). In this context, pSILAC approaches share general advantages, as they target endogenous proteins expressed at physiological levels in untreated cells. We therefore sought to cross-examine our present turnover data with previous studies performed using pSILAC in human (HeLa and Jurkat cells, Boisvert *et al*, 2012; Fierro-Monti *et al*, 2013) and mouse (NIH3T3 cells, Schwanhausser *et al*, 2011). We observed a positive correlation of our data with these other protein stability studies, with Spearman's correlation coefficients of 0.48, 0.44 and 0.49 (Schwanhausser *et al*, 2011; Boisvert *et al*, 2012 and Fierro-Monti *et al*, 2013), respectively. Moreover, comparison of gene ontology enrichment analyses indicated that protein turnover rates seem to be conserved between different cell lines and organisms. Despite similar experimental procedures, there was still a lot of unexplained variation between the data sets, which may be attributed to differential regulation of protein stability in different cell lines or differential growth conditions in the case of the Jurkat cell line, while data processing and interpretation are other factors to consider. Importantly, we clearly demonstrate the contribution of proteoform-specific turnover rates and that differences in stability between N-terminal protein variants may in principle only be resolved by focusing on their unique peptides, whereas shared peptides are a potential source of error in turnover measurements.

Protein turnover measurements were recently extended to protein isoforms and (modified) protein pools (Ahmad *et al*, 2012; Boisvert *et al*, 2012). The Lamond group provided evidence for differential stabilities and localisation of proteoforms. Their shotgun approach, however well suited to study splice variants, had limited application for N-terminal proteoforms. Our pSILAC/N-terminal COFRADIC approach can be further extended to study changes in stability and proteoform expression levels during dynamic processes or in different tissues (Fierro-Monti *et al*, 2013; Kristensen *et al*, 2013). Since targeting to diverse micro-environments allows fine-tuning of local protein concentrations, adding a spatial context to protein turnover may increase our understanding of the functional diversity of proteomes in future studies.

# Materials and Methods

### Cell culture

For N-terminal COFRADIC analyses, human Jurkat T-lymphocytes (clone E6-1 ATCC TIB-152, ATCC) were subjected to SILAC labelling (Ong *et al*, 2002). Cells were cultured in a 5% $CO_2$ gas-equilibrated, humidified incubator in Roswell Park Memorial Institute (RPMI)—1640 medium without arginine and lysine (Silantes, GmbH, Munich, Germany) containing 10% fetal bovine serum dialysed using a

10-kDa cut-off membrane (Gibco, Life Technologies, Paisley, Scotland, UK), 2 mM alanyl-L-glutamine dipeptide (GlutaMAX, Gibco), 50 units/ml of penicillin (Gibco), 50 μg/ml of streptomycin (Gibco) and 146 μg/ml light ($^{12}C_6$) L-lysine (Cambridge Isotope Labs, Andover, MA, USA). Media were supplemented with either light (Arg$^0$, $^{12}C_6^{14}N_4$) or medium (Arg$^6$, $^{13}C_6^{14}N_4$) L-arginine (Cambridge Isotope Labs) at a final concentration of 48 μg/ml at which Arg to Pro conversion was not detected. To achieve a complete incorporation of the labelled amino acids, cells were maintained in culture for seven population doublings (Fig EV5).

The pulsed SILAC strategy for protein turnover measurements was previously described by Boisvert *et al* (2012). Briefly, suspension cultures containing equal numbers of unlabelled (Arg$^0$) and labelled (Arg$^6$) cells were taken and the medium was removed by centrifugation for 5 min at 350 *g*. Cell pellets were washed with PBS and centrifuged for 5 min at 350 *g*. The unlabelled cells were then suspended in light RPMI medium, whereas the medium labelled (Arg$^6$) cells were suspended in medium containing heavy Arg (Arg$^{10}$, $^{13}C_6^{15}N_4$) L-arginine. Following medium change, separate starting cultures of $10^7$ cells were cultured at a concentration of $2 \times 10^5$ cells/ml for periods of 0.5, 1.5, 4, 8, 12, 24 or 48 h, after which cultures of $10^7$ cells were collected by centrifugation. Following two rounds of re-suspending in PBS and centrifugation at room temperature, cell pellets were immediately frozen on dry ice and stored at −80°C.

Proteins were isolated from cell pellets by three rounds of freezing and thawing in 1 ml of 50 mM sodium phosphate buffer, pH 7.5, and cell debris was removed by a 10-min centrifugation at 13,200 *g* at 4°C. The protein concentration in the supernatants was measured by the Bradford method, and for every time point sample, an equal amount of medium/heavy labelled proteome sample was mixed with the corresponding equivalent of light control sample.

### Cycloheximide pulse-chase experiments

100 μg/ml cycloheximide was added to Jurkat cells that were cultured at a density of 180,000 cells/ml. A total of $3.46 \times 10^6$ cells were harvested after 0, 0.5, 1.5, 4, 8, 12 or 24 h of treatment and pelleted for 3 min at 800 g. Cells were lysed in 100 μl of RIPA buffer (50 mM Tris–HCl pH 8.0, 150 mM NaCl, 1% NP-40) with protease inhibitors added (Roche). All lysates were flash-frozen and stored at −80°C until further processing. Samples were thawed and centrifuged for 10 min at 13,200 *g*, followed by measurement of protein content using the Bradford method. Forty micrograms of protein material from each sample was separated by SDS–PAGE (4–12% Criterion Bis-Tris gel, Bio-Rad), and protein levels were monitored by Western blotting. The following antibodies were used: anti-lamin B (NA12, Calbiochem), anti-securin (ab3305, Abcam), anti-β-catenin (C2206, Sigma), anti-SYF2 (ab3610, Abcam), anti-GAPDH (ab9484, Abcam), anti-actin (A2066, Sigma) and anti-α-tubulin (T5168, Sigma). The intensity of bands was assessed using the LICOR Odyssey software for Western Blot image processing.

To validate the proteomics-derived differential protein turnover of dbTIS and aTIS proteoforms of MARE2 and AN32E, complete cDNA clones were obtained from the ORFeome 7.1 collection (ID 4499) and I.M.A.G.E. (ID 3047915), respectively. Site-directed mutagenesis (Stratagene, Agilent Technologies) by mutating ATG (Met)

to TTG (Leu) codons was performed to analyse the expression of selected proteoforms. (Mutagenised) constructs were subcloned into the eukaryotic pEF-DEST51 expression vector (Gateway, Life Technologies) and transfected in HCT116 cells using Fugene HD (Promega) at a 0.8/3/150,000 ratio of DNA [μg], Fugene HD [μl] and number of cells. Cycloheximide pulse-chase experiments were performed as described above for Jurkat cells, except that 150,000 HCT116 cells were harvested for every overexpressed construct and time point analysed. Cells were lysed in 80 μl RIPA buffer, and about 10 μg of total protein material was separated by SDS–PAGE. Degradation of overexpressed proteoforms was monitored by Western blotting using the anti-V5 antibody (R96025, Life Technologies) and compared to the turnover of stable proteins such as actin (Schwanhausser *et al*, 2011).

### N-terminal COFRADIC

The combined proteome mixtures were analysed by N-terminal COFRADIC (Staes *et al*, 2011; Van Damme *et al*, 2011). In summary, proteins are first blocked at their N-ends (α-amines) and lysines (ε-amines) by acetylation. After trypsin digestion, terminal peptides are enriched for by means of SCX at low pH and by applying two consecutive RP-HPLC separations. In the latter, any remaining internal peptides and C-terminal peptides are shifted by reaction of their free α-amino groups (new free α-amino groups introduced by the action of trypsin) with 2,4,6-trinitrobenzenesulphonic acid (TNBS), thereby allowing for their segregation from N-terminal peptides. Fractions enriched for N-terminal peptides are subsequently analysed by means of LC-MS/MS. All steps of the N-terminal COFRADIC analysis, including SCX enrichment, were performed as described (Staes *et al*, 2011). More specifically, solid guanidinium hydrochloride (Gu.HCl) was added to the proteome mixtures to a final concentration of 4 M in order to denature all proteins. Next, proteins were reduced and alkylated using TCEP and iodoacetamide, respectively. Upon desalting of the *S*-alkylated samples in 50 mM sodium phosphate (pH 8.0), primary free amines were *N*-acetylated (no distinction between can be made between *in vivo* Nt-free (and thus *in vitro* Nt-acetylated) and *in vivo* Nt-acetylated N-termini by adding sulfo-N-hydroxysuccinimide (NHS) acetate (Pierce). Twice the molar excess of glycine over the NHS ester was subsequently added to quench any non-reacted NHS-acetate. Possible O-acetylation of Ser, Thr or Tyr residues was reverted by adding hydroxylamine (Fluka) to the modified protein mixtures. A final desalting step was performed in protein digestion buffer (10 mM ammonium bicarbonate, pH 7.9), and the proteomes were digested overnight at 37°C with sequencing-grade, modified trypsin (Promega, Madison, WI, USA) (enzyme/substrate of 1/100, w/w). Subsequent steps of the N-terminal COFRADIC analysis, including SCX enrichment, were performed as described (Stes *et al*, 2014).

Following label swapping, one N-terminal COFRADIC analysis was performed per time point (7 N-terminal COFRADIC analysis in total). During the primary RP-HPLC separation of the actual N-terminal COFRADIC procedure, peptides were collected in 13 fractions of 4 min each. Next, 12 secondary fractions were collected per primary fraction and pooled as described in Staes *et al* (2011), overall resulting in 36 samples for LC-MS/MS analysis per N-terminal COFRADIC set-up.

### LC-MS/MS analysis

LC-MS/MS analysis was performed using an Ultimate 3000 RSLC nano HPLC (Dionex, Amsterdam, the Netherlands) in-line connected to an LTQ Orbitrap Velos mass spectrometer (Thermo Fisher Scientific, Bremen, Germany). The sample mixture was loaded on a trapping column (made in-house, 100 μm I.D. × 20 mm, 5-μm beads C18 Reprosil-HD, Dr. Maisch). After back flushing from the trapping column, the sample was loaded on a reverse-phase column (made in-house, 75 μm I.D. × 150 mm, 5-μm beads C18 Reprosil-HD, Dr. Maisch). Peptides were loaded in solvent A' (0.1% trifluoroacetic acid, 2% acetonitrile (ACN)) and separated with a linear gradient from 2% solvent A'' (0.1% formic acid) to 50% solvent B' (0.1% formic acid and 80% ACN) at a flow rate of 300 nl/min followed by a wash reaching 100% solvent B'. The mass spectrometer was operated in data-dependent mode, automatically switching between MS and MS/MS acquisition for the ten most abundant peaks in a given MS spectrum. Full-scan MS spectra were acquired in the Orbitrap at a target value of 1E6 with a resolution of 60,000. The 10 most intense ions were then isolated for fragmentation in the linear ion trap, with a dynamic exclusion of 20 s. Peptides were fragmented after filling the ion trap at a target value of 1E4 ion counts. Mascot Generic Files were created from the MS/MS data in each LC run using the Mascot Distiller software (version 2.4.2.0, Matrix Science, www.matrixscience.com/Distiller.html). To generate these MS/MS peak lists, grouping of spectra was allowed with a maximum intermediate retention time of 30 s and a maximum intermediate scan count of 5. Grouping was done with a 0.005 Da precursor tolerance. A peak list was only generated when the MS/MS spectrum contained more than 10 peaks. There was no de-isotoping and the relative signal-to-noise limit was set at 2.

The generated MS/MS peak lists were searched with Mascot using the Mascot Daemon interface (version 2.3.01, Matrix Science). Searches were performed in the Swiss-Prot database with taxonomy set to human (UniProtKB/Swiss-Prot database version 2011_08, containing 20,244 human protein entries). The Mascot search parameters were set as follows: acetylation at lysine side chains, carbamidomethylation of cysteine and methionine oxidation to methionine-sulfoxide were set as fixed modifications. Variable modifications were acetylation of N-termini and pyroglutamate formation of N-terminal glutamine (both at peptide level). Endoproteinase semi-Arg-C/P (semi-Arg-C specificity with Arg-Pro cleavage allowed) was set as enzyme allowing for no missed cleavages. Mass tolerance was set to 10 ppm on the precursor ion and to 0.5 Da on fragment ions. Peptide charge was set to 1+, 2+, 3+ and instrument setting was put to ESI-TRAP. Only peptides that scored above the threshold score, set at 99% confidence, were withheld. The FDR was estimated by searching a decoy database (a shuffled version of the human Swiss-Prot database version 2011_08) and ranged from 0.32% to 2.27% for individual samples, with a global FDR of 1.18% on the spectrum level.

### Ribosome profiling

For ribosome profiling, Jurkat cells were grown in RPMI medium (Gibco) supplemented with 10% fetal bovine serum, 2 mM alanyl-L-glutamine dipeptide (GlutaMAX, Gibco), 50 units/ml penicillin and 50 μg/ml streptomycin at 37°C and 5% $CO_2$. Cultures were treated

with 50 µM lactimidomycin (LTM) (Ju *et al*, 2005; Schneider-Poetsch *et al*, 2010) or 100 µg/ml cycloheximide (CHX) (Sigma, USA) for 30 min at 37°C before cell harvest. Subsequently, cells were collected by centrifugation (5 min at 300 *g*), rinsed with ice-cold PBS and recovered again by centrifugation in the presence of CHX. $10^8$ cells were re-suspended in 1 ml ice-cold lysis buffer [10 mM Tris–HCl, pH 7.4, 5 mM $MgCl_2$, 100 mM KCl, 1% Triton X-100, 2 mM dithiothreitol (DTT), 100 µg/ml CHX, 1× complete and EDTA-free protease inhibitor cocktail (Roche)] (Guo *et al*, 2010). Following 10 min of incubation on ice, samples were passed through QIAshredder spin columns (Qiagen). The flow-through was clarified by centrifugation for 10 min at 16,000 × *g* and 4°C. Six hundred microlitres of the recovered supernatant was subjected to RNase I (Life Technologies) digestion using 1000 U of enzyme. Digestion of polysomes proceeded for 55 min at room temperature and was stopped with SUPERase.In RNase Inhibitor (Life Technologies). Next, monosomes were recovered by ultracentrifugation at 75,000 rpm in a cooled TLA-120.2 rotor over a 1 M sucrose cushion in 20 mM HEPES-KOH pH 7.4, 5 mM $MgCl_2$, 100 mM KCl, 2 mM DTT, 100 µg/ml CHX and 100 U/ml Superase.In. Subsequent steps were performed as described by Ingolia *et al* (2011) with some minor adjustments. RNA was extracted from the samples using a heated acid phenol– chloroform– isoamyl alcohol (125:24:1) procedure. Seventeen micrograms of RNA was subjected to electrophoresis under denaturing conditions in 15% TBE-Urea poly-acrylamide gel (Life Technologies). Ribosome-protected fragments (RPFs) of 28–34 nucleotides were extracted from the gel in RNase-free water for 10 min at 70°C and precipitated with 1/10 volume of 3 M sodium acetate pH 5.2, 1.5 µl GlycoBlue (Life Technologies) and 1 volume of isopropanol overnight at −80°C. Following a dephosphorylation reaction (end-repair) with 10 U of T4 polynu-cleotide kinase (New England Biolabs), the RPFs were depleted of ribosomal RNA contaminants using the Ribo-Zero Magnetic Gold Kit (Illumina) according to the manufacturer's protocol. Subsequently, fragments were ligated to a 1.5 µg of Universal cloning linker using 200 U of T4 RNA ligase II (both New England Biolabs). The ligated product was size-selected using denaturing gel electrophoresis and purified from gel as described above. Subsequently, reverse tran-scription (RT) was performed with 200 U of SuperScript III (Life Technologies) according to Ingolia *et al* (2011), followed by a size selection of the RT product (described above). Next, samples were subjected to circular ligation using 100 U of CircLigase (Illumina) and amplified by PCR using Phusion polymerase (New England Biolabs) for 12 cycles of denaturation (10 s at 98°C), annealing (10 s of 65°C) and elongation (5 s at 72°C) using primer pairs compatible with the Illumina sequencing platform (Ingolia *et al*, 2011). The resulting ribosome profiling libraries of LTM- and CHX-treated Jurkat cells were sequenced on a NextSeq 500 instrument (Illumina) to yield 75-bp single-end reads.

## Measurement of proteoform turnover

Arg-ending N-terminal peptides were selected that started at posi-tion 1 or 2 (entries stored in the Swiss-Prot database) or derived from alternative translation initiation at positions > 2) and which complied with rules of iMet processing and Nt-acetylation (Van Damme *et al*, 2014). For every unique N-terminus (unique proteoform), SILAC ratios (M/L, H/L and H/M) calculated for the

highest scoring peptide identification were included in the analysis. M/L and H/L ratios were normalised, according to equation (1):

$$\text{if } M + H = L \text{ then } M/L + H/L = 1 \tag{1}$$

where L, M and H denote light, medium and heavy isotope-labelled peptide intensities.

M/L and H/L ratios were plotted in function of time as degrada-tion and synthesis curves, respectively. The crossing over point between the two curves corresponds to 50% protein turnover, see Equation (2):

$$\text{if } M/L + H/L = 1 (\text{Eq. 1}) \text{ and } M/L = H/L \text{ then } M/L = H/L = 1/2 \tag{2}$$

Fifty per cent protein turnover was calculated as the crossing point between protein degradation and synthesis curves for proteo-forms quantified in at least three time points. We assumed that protein degradation follows a first-order process. Therefore, an exponential curve with one parameter (B) was fitted to M/L data using the R software (version 0.98.739) and the turnover time was calculated as follows:

$$M/L = e^{\left(-k_{\text{deg}} - \frac{\ln(2)}{t_{cc}}\right)t}, \text{ simplified to } M/L = e^{Bt}, \text{ hence } t_{\text{turn}} = \frac{\ln(1/2)}{B} \tag{3}$$

where $k_{\text{deg}}$ is the protein degradation constant; $t_{cc}$, cell doubling time; and $t_{\text{turn}}$, turnover time.

Alternatively, the turnover time was approximated from linear equations between two M/L and two H/L points lying on opposite sides of the crossing point. We observed a very high correlation between the exponential and the linear method (Pearson coefficient of 0.98). Exponential curve fitting was used for protein turnover determination if the quality of fit determined by the coefficient of determination ($R^2$) was at least 0.8. In other cases, turnover time was calculated from the linear approximation, provided a descending trend of M/L changes in time was observed. Turnover measurements fulfilling criteria described above were regarded as valid, while other turnover rates remained unassigned and were excluded from analysis. Protein degradation constants were derived from the parameter B (Equation 3) of individual M/L profiles, and protein half-lives ($t_{1/2}$) were calculated accordingly to equation (4):

$$t_{1/2} = \frac{\ln(1/2)}{-k_{\text{deg}}} = \frac{\ln(1/2)}{\left(B + \frac{\ln(2)}{t_{cc}}\right)} \tag{4}$$

Due to the constrains of equations (3) and (4), negative values of $k_{\text{deg}}$ were disregarded and protein half-lives estimated to be infinite due to continuous cell divisions: $t_{1/2} \gg t_{cc} \rightarrow t_{1/2} \approx \infty$.

## Quantification of proteoform abundance

Per unique proteoform, spectral counts were averaged among all time points whenever identified (using an arithmetical mean). Normalised spectral abundance factor (NSAF) values (Paoletti *et al*, 2006) were calculated based on amino acid lengths of individual

proteoforms. All proteoforms identified in at least one time point were taken as the complete data set (Table EV1A).

## Analysis of ribosome profiling data

Ribosome profiling (Ribo-Seq) performed in Jurkat cells treated with CHX to study translation elongation and LTM to study translation initiation were analysed using the PROTEOFORMER pipeline (Crappe *et al*, 2015). PROTEOFORMER was used to perform the quality control, mapping onto a reference genome, identification of TIS and generation of a protein database.

The 3′ adapter sequences were clipped using the fastx_clipper, and reads shorter than 26 or longer than 34 nucleotides were discarded for the Ribo-Seq-based TIS calling. The  reads were first mapped onto small nuclear RNA, tRNA and rRNA sequences to eliminate contaminating reads. The remaining reads were subsequently mapped onto the human GRCh37 reference genome (Ensembl annotation bundle 75) allowing for a maximum of two mismatches along the entire read. Reads mapping to a maximum of 16 locations on the genome were allowed and retained for every loci. All mapping steps were performed using STAR 2.4.0i. RPF alignments were reduced to a specific position (the P-site nucleotide) using an offset from the 5′ end of the alignment based on the read length (+12, +13 and +14, respectively, for alignment of $\leq 30$ bases long, 31–33 bases long and $\geq 34$ bases long). Translation start sites were identified by setting different parameter values for the different TIS categories in the tool. A TIS was called at the Ensembl-annotated start site if (i) it had a minimum read count of 5; (ii) the position was a local maximum within a 7-nucleotide window (one codon upstream and one codon downstream); (iii) the difference between the LTM (foreground) and CHX (background) normalised ratio ($R_{LTM-CHX}$) within the window reached at least 0.01. A TIS was called within the coding region of a transcript (downstream of annotated start sites) if it had a minimum read count of 15 and $R_{LTM-CHX}$ of 0.15. For TIS located at the 5′UTR, 3′UTR and other non-protein-coding transcripts, the minimum read count was set to 10 with a $R_{LTM-CHX}$ of 0.05.

We used the above settings to define translation initiation events in Jurkat cells. Taking into consideration every annotated transcript splice variant, all possible ORFs starting at the detected TIS sites were *in silico* translated. The N-terminal peptides identified in our pSILAC experiment ($N = 2,578$) were mapped onto the Ribo-Seq-derived protein sequences allowing to associate peptides with ORFs and their corresponding transcripts.

## Analysis of translation initiation and splicing

To study the contribution of splicing and (alternative) translation initiation to N-terminal proteoform expression, we mapped the N-terminal peptides identified in our proteomics experiment onto the human Swiss-Prot Isoform, UniProt TrEMBL, Ensembl human protein sequences next the Ribo-Seq data from Jurkat cells. Further, a comparison was made with Ribo-Seq data obtained from another human cell line (HCT116 colorectal cancer cells, unpublished data).

In case of Ribo-Seq matching evidence, a peptide corresponding to position 1 was assumed to be generated via translation initiation. A peptide mapping at position 2 was considered a TIS only if the first methionine was preceded by an alanine, cysteine, glycine,

proline, threonine or valine and therefore likely co-translationally processed. We used the Jurkat data as reference (i) to retrieve $R_{LTM-CHX}$ values for called TIS sites; (ii) to associate transcripts with a normalised RPF read count (the normalised read count was calculated by summing all the reads that map across the transcript and dividing by its length) and iii) to determine their exon coverage. Since an N-terminal peptide might map to multiple transcripts, additional information regarding the expression was provided for every matching transcript to allow for a discrimination of transcripts poorly supported by the Ribo-Seq data.

We further investigated whether a downstream peptide is pointing to a possible annotated splicing or alternative translation event by checking whether this peptide matched the N-terminal protein sequence previously reported in the protein databases: Ensembl, Swiss-Prot Isoform or UniProt TrEMBL protein sequences. A peptide was considered as a possible N-terminus of a splice variant if it mapped at position 1 or 2 to a given sequence in Ensembl or Swiss-Prot Isoform databases. Further, we qualified the Ensembl-derived evidence of splicing as either poor (less likely) or good (likely) splicing events. A downstream peptide was considered to be supported by a likely splicing event if any of the Ensembl protein sequences associated with the peptide had a transcript support level less than or equal to three (with one being the highest rank). If none of the protein sequences associated with the peptide had a transcript support level less than three, it was considered as a poorly supported splicing event or not applicable if there was no transcript support level assigned to any of the valid transcripts. For N-terminal peptides matching an N-terminus of a Swiss-Prot Isoform sequence, we further retrieved the information if this protein isoform is generated via splicing or alternative translation initiation. The above-mentioned data are presented in Table EV1C.

## Analysis of proteoform turnover rates in macromolecular complexes

To investigate the dynamic properties of macromolecular complex substituents, protein turnover times were mapped to members of human protein complexes defined by the CORUM database (core set of 1,343 human complexes, Ruepp *et al*, 2008). Only Swiss-Prot-annotated proteoforms (dbTIS) with the most plausible iMet processing status (Van Damme *et al*, 2011) and high-quality turnover measurement were considered (Arnesen *et al*, 2009; Van Damme *et al*, 2011). Absolute differences in turnover were calculated pairwise between all members of a complex. Subsequently, one-sided Mann–Whitney *U*-test at a 95% confidence interval (R, wilcoxon.test, stats package) was applied to detect proteins with significantly higher absolute differences than the remaining members of a complex.

## Gene ontology enrichment analysis

Swiss-Prot accessions of the 1,894 proteoforms with high-quality measurement of stability of one representative N-terminal proteoform were ranked based on their corresponding turnover values. Subsequently, an enrichment analysis of GO terms was performed using GOrilla (http://cbl-gorilla.cs.technion.ac.il/; Eden *et al*, 2009) using the "single ranked list of genes" option. Duplicate accessions

were removed keeping the highest ranking instance per accession. The enrichment *P*-values were computed using mHG (minimal Hyper Geometric) statistics and corrected for multiple testing using the Benjamini and Hochberg method giving the FDR *q*-values. GO terms enriched in stable or unstable proteins with FDR *q*-values ≤ 0.05 were subsequently summarised using REViGO (Supek *et al*, 2011). Redundant GO terms were removed and the remaining terms visualised in bar charts according to their significance (−log$_{10}$(*q*-value)). Analogously, we performed an enrichment analysis of GO terms in aTIS vs. dbTIS proteoforms using the GOrilla "target and background" option.

### Prediction of protein intrinsic disorder

Canonical protein sequences identified in at least three time points (1,972 proteoforms) were retrieved and adjusted to start at the initiating residue of the N-terminal peptide identified. Of these proteoforms, the percentage of intrinsically disorder was predicted using RAPID (http://biomine-ws.ece.ualberta.ca/RAPID/index.php; Yan *et al*, 2013) which implements the IUPred algorithms (Dosztanyi *et al*, 2005a,b).

### Hydropathy of protein N-termini

The grand average of hydropathy (GRAVY) was calculated for the 2- and 10-amino acid-long N-terminal sequence of every proteoform (1,894 proteoforms) at http://www.gravy-calculator.de/. Increasing positive score indicated greater hydrophobicity. The calculation was based on the sum of hydropathy values of amino acids in Kyte–Doolittle scale (Kyte & Doolittle, 1982), divided by the number of residues. Correlation between GRAVY values and turnover times was estimated using Pearson and Spearman correlation coefficients.

### Statistical analysis of turnover, abundance and disorder of aTIS versus dbTIS

Differences in turnover, abundance and disorder values between aTIS and dbTIS groups were measured for paired dbTIS-aTIS data using a Wilcoxon signed-rank one-sided test for paired samples (R, stats package) and independently between the whole aTIS and dbTIS groups using a Kruskal–Wallis test for difference in distribution (R, stats package), multiple comparison test after Kruskal–Wallis (R, pgirmess package) and Mann–Whitney one-sided test for independent samples. All analyses were performed in R at *P* ≤ 0.05 significance.

### Category enrichment analysis in turnover, disorder and abundance data

1D enrichment analysis of Pfam, KEGG, GO slim and other categories was performed using the Perseus software (Cox & Mann, 2012) at *P* ≤ 0.01 significance with a two-sided test with Benjamini–Hochberg FDR correction. Turnover, disorder and abundance values corresponding to each categorical term were tested for deviations from the complete distribution, and a preference for larger or smaller values was assessed independently for every variable. Correlation of turnover, disorder and abundance values was assessed using Spearman rank correlation.

### Relation between protein turnover and N-terminal amino acid composition

All proteoforms with measured turnover rates (including proteoforms with alternative iMet processing; 1,941 in total) were categorised according to the identity of their utmost N-terminal residues. Additionally, N-termini susceptible to iMet processing were further categorised as either Met retaining or Met cleaved. The general impact of N-terminal residues and iMet processing (categorical variables) on protein turnover was assessed by a Kruskal–Wallis rank sum test (R, stats package). A subsequent multiple comparison test for Kruskal–Wallis (R, pgirmess package) and pairwise Wilcoxon rank sum test (without correction for multiple testing; R, stats package) allowed to indicate the most deviating subgroups (with *P* < 0.05).

### Data availability

MS/MS data was converted using PRIDE Converter (Barsnes *et al*, 2009) and is available through the PRIDE database (Martens *et al*, 2005) with the data set identifier PXD002091 and DOI: 10.6019/PXD002091.

Ribo-Seq sequencing data has been deposited in NCBI's Gene Expression Omnibus (Edgar *et al*, 2002) and is accessible through GEO Series accession number GSE74279 (http://www.ncbi.nlm.nih.gov/geo/query/acc.cgi?acc = GSE74279).

**Expanded View** for this article is available online.

## Acknowledgements

DG is supported by a Ph.D. grant of the Institute for the Promotion of Innovation through Science and Technology in Flanders (IWT-Vlaanderen). PVD and KG acknowledge support from the Research Foundation-Flanders (FWO-Vlaanderen), respectively, project numbers G.0269.13N and G.0440.10.

## Author contributions

DG performed experiments, analysed data, drafted and revised the manuscript. EN analysed data. KG drafted and revised the manuscript. PVD conceived the study, performed the proteomics experiments, analysed data, drafted and revised the manuscript.

## Conflict of interest

The authors declare that they have no conflict of interest.

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
