## [Review Process File · Molecular Systems Biology]

Positional proteomics reveals differences in N-terminal proteoform stability

Daria Gawron, Elvis Ndah, Kris Gevaert and Petra Van Damme

Corresponding author: Petra Van Damme, Ghent University

Review timeline:

Submission date:	02 April 2015
Editorial Decision:	28 June 2015
Resubmission:	27 October 2015
Editorial Decision:	22 December 2015
Revision received:	07 January 2016
Accepted:	20 January 2016

Transaction Report:

1st Editorial Decision

28 June 2015

The reviewers find the general topic of your study of potential interest. The three reviewers all raise however significant concerns with regard to essential aspects of the study, including the verification of the accuracy of the data and the rigour of the analysis, the validity of major conclusions with regard to the biological relevance of the measured turnover rates. As such, while the approach seems interesting, the study appears underdeveloped and addressing all the points raised by the reviewers may extend beyond the scope and timeframe of a normal revision.

As such, and given the limited level of support expressed by the present reviewers, we have no choice but to return the study with the message that we cannot publish it.

Nevertheless, we acknowledge that the reviewers expressed interest in the subject matter and your approach. We would therefore not be opposed to consider the submission of a new and extended study, provided the following aspects can be convincingly and carefully addressed with suitable additional analyses and data:

- To convince the readers of the robustness and accuracy of the data, the assumptions made during the data analysis should be rigorously justified and a more detailed assessment of the accuracy of the measured turnover rates should be performed, including a more convincing validation.
- Further in-depth analyses by comparison with existing data are required to clarify the contribution of alternative splicing, alternative translation initiation and intracellular proteolytic processing to the diversity in N-termini observed.

- Additional experimental verification and accurate measurements are required to ascertain the conclusions with regard to the N-end rule and effect of Nt-acetylation.
 - The impact of proteoforms on protein complex should also be supported by further analysis and data to be fully conclusive.
-

Reviewer #1:

Summary

In the current manuscript Gawron and colleagues used a proteomics approach to specifically enrich for N-terminal peptides (termed: COFRADIC). Specifically, human Jurkat cells were pulse labeled by a SILAC approach (pSILAC) and coupled to quantitative LC-MS analyses to obtain a systems-level overview of protein stability of N-terminal proteoforms (7 distinct time-points were sampled). Overall, turnover rates and rough abundance values were reported for over 2,500 proteoforms (based on LC-MS of the N-terminal peptides). A number of high-level conclusions are drawn from these data such as, GO enrichment of stable vs. unstable proteins; abundance and turnover rates of N-terminal proteoforms, including for peptides from alternative translation initiation; proteoforms and their impact on protein complex stability, etc.

General remarks

Overall I found this paper interesting. It provided a mile high view of expression, abundance and stability/turnover of proteins (and N-terminal proteoforms) using a modern proteomics technology and a previously published approach COFRADIC-pSILAC. I found it a clever way of obtaining these types of information (on a global level) that are likely not obtainable using classic approaches such as 35S pulse-chase. Hence, these data provide a conceptual advance to our understanding of protein stabilities, turnover rates and expression of proteoforms from alternative translation initiation. The paper was generally well written with a somewhat technical focus. Additional validations or discussion of selected examples could certainly boost the applicability to a broader biology audience. The section of macromolecular complexes was highly speculative and rather hard to judge a present time (see below).

General concerns

1) Page 7: "...From the total of 2,578 N-terminal peptides identified we retained 1,972 peptides, pointing to 1,637 dbTIS and 335 aTIS, that were identified in at least 3 of the time points analysed...". If I understand this sentence correct, ~600 peptides were removed, since they were identified in less than 3 time-points. The other peptides were used (S1 A)? It is very hard to gather these data from looking at this supplemental table.

a. There is quite a bit of missing data in this table, possible due to random sampling. Have the authors taken a closer look at what happened to these peptides and potentially if some of this data could be rescued by matching between runs?

2) Page 8: what is the rationale of using a p-value cut-off of 0.001?

a. I found the GO enrichment figure hard to read. Wouldn't some type of bar chart be much cleaner?

3) Page 10: It wasn't completely clear how the authors used the spectral counts/NSAF to compare abundance of aTIS vs. dbTIS?

a. Was this done only for the 100 dbTIS/aTIS pairs?

b. Only spectral counts for the two distinct N-terminal peptides were used, and from what I can see in SI Table 1A these were somehow averaged over all time-points? The majority of these counts are quite low <10, arguing against any useful quantitative data.

c. In this context, how do the authors know that different N-terminal peptides (proteoforms) have the identical recovery during the COFRADIC enrichment? Have they ever performed any type of recovery experiments using synthetic peptides?

4) The data on protein complexes was interesting and intuitively it makes sense that members of a protein complex have correlated turnover rates. The part of alternative proteoforms and their impact on complex stability was completely unclear and to my mind speculative at best. Unless the authors can show that an alternative proteoform gets incorporated into the same protein complex and as a result affects the turnover rate of the entire complex, which I couldn't find anywhere in the manuscript, then this part of the paper should be toned down or even removed.

5) While it was nice to see at least a little bit of validation (Figure 7) I found the over-expression studies comparing turnover for dbTIS vs aTIS proteins somewhat crude. To my mind a targeted proteomics assay (MRM) would have been tailor made for these experiments and significantly more accurate.

Minor points

1) Page 5: This reviewer was not completely certain what the authors meant by this sentence "...Important here is that we rely on known N-terminal modifications to unambiguously assign an identified peptide as a proxy for a protein's N-terminus..."?

Reviewer #2:

MSB 15-6215

Positional proteomics reveals differences in N-terminal proteoform stability

Gawron et al.

Synopsis

The authors utilize a double SILAC labeling of Jurkat cells combined with COFRADIC (a technique to isolate N-terminal peptides) to begin to study how proteins with different N-term amino acids, or those produced from different translation start sites, differ in their half lives. They report that the extent of intrinsic disorder was correlated with turnover times.

Minor problems

1. Many experimental details are missing here. For example, it would be helpful for those unfamiliar with the COFRADIC method to at least provide a brief description of how it works, rather than simply referring to previously published protocols.

2. The authors do not include sufficient information on how the mass spectrometry was conducted; e.g. what was the resolving power of the orbitrap for MS1? What was the cutoff TIC for taking an MS2? Do each of these analyses represent a single mass spec run? How many mass spec runs were conducted in total? etc. This information must be included to properly judge this work. I would also have really liked to have seen examples of the spectra in the main text, especially for the peptides used for evidence of "alternative" translation initiation sites. On a related note, the use of a single database search engine, and data presented without any kind of statistical analysis, would be considered somewhat out-of-date by most mass spectrometry professionals (and specialized mass spec journals), at this point in time. A high quality

mass spectrometry analysis would now utilize at least two different search engines, followed by a rigorous statistical analysis to assign a confidence value for each peptide and protein identification (e.g. Peptide and Protein Prophet or similar tools).

Major concerns

1. The authors indicate on page 7 that the N-terminal peptides reported here were observed in at least 3 time points. Much more detail should be provided here: e.g. I am not convinced that an accurate half-life can be derived when a peptide was detected at times 0.5, 1.5 and 48 hrs. How many such cases are in this dataset, and how did the authors deal with such cases? Is there a statistical argument to be made that this number of observations is sufficient to estimate a proper half-life? Can the authors demonstrate that this is so, using a well-characterized protein as an example? I see this as a significant weakness here, unless the authors can make a convincing argument otherwise.

On a related note, the authors should at least also discuss the possibility that additional post-translational modifications of the N-terminal peptides (phosphorylation, etc.) could render them undetectable by mass spectrometry: in these cases, the apparent half-life would be short, but not reflect the true half-life of the protein. One additional possibility that should also be discussed here - Do different N-termini contain different numbers of lysine residues that could be ubiquitinated and therefore affect protein half-life?

2. Most translation start sites have been predicted *in silico*, usually by the presence of a Kozak sequence in a predicted (or experimentally observed) mRNA. This is not even mentioned in the manuscript, but definitely should be for the audience to have a proper understanding of how proteins may end up with different N-termini. The authors suggest that, "...leaky ribosome scanning is likely to be responsible for the majority of downstream translation events observed", but this is not necessarily true - at all. What proportion of the N-terminal peptides identified here could be due to alternative splicing of 5' exons, such that the predicted upstream Met codon is not even present in the resulting mRNA? This would be critical for really understanding what is going on here. How many of the aTIS would be predicted to be the "proper" start site in ESTs (there are huge databases that could be searched for the presence of such examples)? Moreover, given that 2/3 of the observed "aTIS" peptides were detected in the absence of their canonical counterparts, the authors must at least estimate how many of these "alternative" translation start sites might actually be the proper (but mis-annotated) start sites. (i.e. What proportion of these peptides truly represent "alternative" start site usage?) One additional very useful metric to report would be - How many upstream N-terminal peptides have been previously observed in proteomics studies? (This can be easily addressed by searching the many large proteomics databases, freely available.)

In short, while the authors make some possibly interesting observations, they have not done a good job of exploring/explaining why the aTIS may be present in the first place.

3. Similarly, the authors do a very poor job of relating their findings to previously published papers. e.g. How does the observed half-life of 21.6 hrs compare to previously reported average half-lives of proteins in mammals and model organisms? How do these data on GO categories compare to that previously observed in mammals and other organisms (e.g. quite a lot of protein turnover work has been conducted on budding yeast, but is not even mentioned here)?

4. While the introduction provides an overview of the importance of N-terminal acetylation (on Met or other residues, when the initiator Met is removed), it makes no mention at all of the N-end rule. Several decades of research on how the N-terminal residue of a protein influences its half-life (conducted primarily by Varshavsky and colleagues) is not even brought up here.

It is mentioned later on in the manuscript, where the authors go on to argue (with a small, and rather poorly described dataset) that they see no evidence for N-end rule function in their data. In this reviewer's opinion, this could be the most important observation in their manuscript, but is not backed up with a proper, deep analysis of the evidence (i.e. extraordinarily good data would be required to make this type of extraordinary claim). If their data do not agree with N-end rule predictions, why not?

5. Finally, I find the follow-up "verification" to be quite minimal and not very convincing. The authors simply conduct a few Western blotting experiments of endogenous proteins. These blots are not all of high quality; the Westerns have no molecular mass markers; and I would also like to see the entire blot. More importantly, in order to convince this reviewer that the different proteoforms have significant differences in half-life, several examples would have to be engineered for expression in cells (perhaps with C-terminal epitope tags) and proper half-life experiments conducted.

Reviewer #3:

This is an interesting manuscript that sets out to assess the effect of N-terminal proteoform variation on intracellular stability. The authors use COFRADIC to isolate N-terminal peptides that have been labeled in rather complicated dynamic SILAC experiment.

However, the manuscript actually strays from the implications of the title, and it is really a study of the intracellular stability of proteins in the context of the N-terminus that can be recovered by COFRADIC. Most of the peptides are 'expected' N-terminii. It is predominantly therefore an 'actual' N-terminal profile informed turnover analysis.

There are some issues that should be considered. However, with attention and a more critical analysis, the manuscript could become much better, and worthy of publication.

First, the authors use half-life throughout as the measured parameter. This is fundamentally incorrect. The linear parameter is the first order rate constant, and the half-life is a reciprocal term that distorts the error structure and distribution of the data. The data need to be reexamined in linear parameter space - i.e. the first order rate constant.

To assume a zero order synthesis/first order degradation model (simplistic, but useable) it is important to demonstrate that the loss of isotope follows monoexponential kinetics, and the lack of primary data in the manuscript, or in a supplementary file (why not include an isotope profile, time profile for each protein?).

For the N-terminally-informed turnover study, the identity of the actual N-terminus is one piece of information that is readily decoupled from the rate of degradation. Thus, the study could have used unfractionated analyses to recover k_{deg} - this would have yielded more peptides and more time points, for each protein. The lack of primary data makes it impossible to assess the accuracy with which t_{05} (really, k_{deg}) have been determined.

Secondly, I failed to find a full discussion of the fact that the protein pool used to create the COFRADIC data set is a broken cell preparation. We thus have a mixture of proteins that were destined to end up in different subcellular compartments. Thus, one might posit that the N-end rule only applies to cytosolic proteins.

Further, N-terminal proteolytic events are common processing events even intracellularly. It is important to clarify whether any of the neo-termini could be due to proteolysis as opposed to alternative splicing. How many of the alternative termini were anticipated based on RNAseq analyses?

The obligatory N terminal acetylation does of course converge the acetylated and non-acetylated proteins into one, consistently behaving pool. However, it would have been helpful to repeat the analysis with ¹³C-acetic anhydride, to be able to discriminate the natural from the chemical modifications. These data could then be combined with the turnover data, knowing which was naturally acetylated.

A smaller part of the data set refer to alternative N-terminal proteoforms (assumed to be due to alternative splicing, but other routes to neo-termini?) This is a very interesting part of the work, but it is a minor part of

the work. The turnover study alone is not a hugely original study, even if driven by a single peptide/n-terminal approach. The proteoform segment would require much more analysis, and ideally, a stronger indication of the quality of the turnover rates. For this subset, accurate measure of kdeg, with multiple peptides, should be quite feasible.

Technical points.

It is important to be sure that the labeling protocols work (100% label, complete transition to unlabelling etc.). Some supplementary data affirming this would be helpful. Some indication of the quality of the COFARDIC protocol is needed - how many internal peptides were present in the sample?

I think the doubling time of the Jurkat cells should have been cited. If it is 20h approx. (typical) then this means that most proteins are actually replaced by dilution, not by degradation. Figure 3c should have the doubling time (in h) marked clearly on the graph.

I have some nervousness about the contention that incubating a cell in cycloheximide for 48h would result in a living cell that maintained the machinery and energetics for sustained degradation. Perhaps the authors would address this. They are not dying at 48h, they are dead!

By definition, all peptides are 'one hit wonders'. It is thus critical that we be given a measure of peptide quality. Other journals would request the MSMS spectrum of each of these peptides.

The authors should have included a text file or excel sheet of protein, peptide, peptide quality/score, and kdeg.

Resubmission

27 October 2015

Reviewer #1:

Summary

In the current manuscript Gawron and colleagues used a proteomics approach to specifically enrich for N-terminal peptides (termed: COFRADIC). Specifically, human Jurkat cells were pulse labeled by a SILAC approach (pSILAC) and coupled to quantitative LC-MS analyses to obtain a systems-level overview of protein stability of N-terminal proteoforms (7 distinct time-points were sampled). Overall, turnover rates and rough abundance values were reported for over 2,500 proteoforms (based on LC-MS of the N-terminal peptides). A number of high-level conclusions are drawn from these data such as, GO enrichment of stable vs. unstable proteins; abundance and turnover rates of N-terminal proteoforms, including for peptides from alternative translation initiation; proteoforms and their impact on protein complex stability, etc.

General remarks

Overall I found this paper interesting. It provided a mile high view of expression, abundance and stability/turnover of proteins (and N-terminal proteoforms) using a modern proteomics technology and a previously published approach COFRADIC-pSILAC. I found it a clever way of obtaining these types of information (on a global level) that are likely not obtainable using classic approaches such as 35S pulse-chase. Hence, these data provide a conceptual advance to our understanding of protein stabilities, turnover rates and expression of proteoforms from alternative translation initiation. The paper was generally well written with a somewhat technical focus. Additional validations or discussion of selected examples could certainly boost the applicability to a broader biology audience. The section of macromolecular complexes was highly speculative and rather hard to judge a present time (see below).

We would like to thank the reviewer for her/his kind appreciation for our work. Next to removing the somewhat speculative analyses on the impact of N-terminal proteoforms on protein complexation, we validated the expression of N-terminal proteoforms by means of ribosome profiling next to tracking their origin of creation (i.e. (alternative) translation initiation and (alternative) splicing), and included an additional discussion of selected examples (see below).

General concerns

1) Page 7: "...From the total of 2,578 N-terminal peptides identified we retained 1,972 peptides, pointing to 1,637 dbTIS and 335 aTIS, that were identified in at least 3 of the time points analysed...". If I understand this sentence correct, ~600 peptides were removed, since they were identified in less than 3 time-points. The other peptides were used (S1 A)? It is very hard to gather these data from looking at this supplemental table.

In supplemental Table S1.A (now renamed to Table EV1A), all 2,578 N-terminal peptides identified are reported together with the number of proteome samples (i.e. coming from the 7 distinct time points analysed) they were identified in (column "count.time.points"). For clarity, the subset of N-termini identified in at least 3 time points is presented in Table S1.B (now renamed to Table EV1B), with additional information such as the expected proteoform these peptides denote (dbTIS or aTIS). As the reviewer correctly points out, 606 peptides were excluded from further downstream analysis due to insufficient sampling (i.e. N-termini identified in less than 3 out of the 7 time points analysed). Legends have now been included holding detailed information regarding the supplementary table content.

a. There is quite a bit of missing data in this table, possible due to random sampling. Have the authors taken a closer look at what happened to these peptides and potentially if some of this data could be rescued by matching between runs?

Originally, we did not use "match between runs" feature as, intrinsic to the N-terminal COFRADIC analysis, each N-terminome comprises the analysis of 36 individual samples by LC-MS/MS. Hence, given the large number of analyses needed (252 analyses in total or over 10 days of LC-MS/MS analysis) and the inherent irreproducibility of the chromatography separations over such a long time, we had low expectations to recover peptide data using "match between runs". Further, at least in our hands, such an extrapolation of peptide identifications based on mass precision and retention time works only optimal when analysing non-fractionated samples.

Since we previously published an extensive catalogue of N-terminal proteoforms in human and mouse (Van Damme et al, 2014), in this study we focused more on the determination of the cellular stability of proteoforms. Although Mascot Distiller doesn't support the "match between runs" which may result in more missing values, it has certain advantages for peptide quantification over other search algorithms such as MaxQuant. More specifically, Mascot Distiller considers the similarity between experimental and theoretical MS spectra, thereby providing an quantification quality indication. Finally, while the union of peptide identifications from multiple search engines can increase the sensitivity of proteomics enquiries (Yu et al, 2010), quantification values yielded by different algorithms are poorly correlated and cannot be directly combined (Colaert et al, 2011). Instead, the variance in ratio values is inconsistent and depends on MS spectrum intensity, which implies multiple normalization steps (Colaert et al, 2011). Since a reliable quantification was essential in our study, the possible benefit of increased number of peptide identifications was considered insufficient to attempt the integration of Mascot and MaxQuant (or other alternative search algorithms) results. Finally, we show that sampling of 3 data points per N-termini are sufficient to reproduce the original 50% turnover time measured for 7 points (See major comment 1, reviewer 2).

2) Page 8: what is the rationale of using a p-value cut-off of 0.001?

a. I found the GO enrichment figure hard to read. Wouldn't some type of bar chart be much cleaner?

The threshold value of 0.001 is applied to the p-values prior to the multiple hypothesis correction on the number of tested GO terms. A stringent cut-off for the p-value (0.001) usually corresponds to a less stringent cut-off of 0.05 at the corrected false discovery rate (FDR) q-values but may deviate slightly, depending on the number of GO terms used for a particular analysis. To improve clarity, we replaced the p-values with corrected FDR q-values using a threshold of 0.05.

As suggested by the reviewer, we modified the GO enrichment figure (see below).

Figure 4 | Gene ontology (GO) term enrichment analysis for unstable (A) and stable (B) proteins. Horizontal bar chart representations are given of significantly enriched GO terms in the human proteome (FDR q -value ≤ 0.05).

3) Page 10: It wasn't completely clear how the authors used the spectral counts/NSAF to compare abundance of aTIS vs. dbTIS?

As indicated in the Materials and Methods section entitled 'Quantification of proteoform abundance', an average of spectral counts per proteoform was calculated among all time points in which the proteoform was identified. Normalized spectral abundance factor (NSAF) values were calculated according to (Paoletti et al, 2006) based on amino acid length of individual proteoforms and the total number of N-terminal proteoform specific spectral counts obtained in our study. To clarify, we now additionally indicated that an arithmetic mean method of averaging was used.

a. Was this done only for the 100 dbTIS/aTIS pairs

First, we performed a global comparison of abundance measures between the entire aTIS (323) and dbTIS (1,571) groups (from a total of 1,894 proteoforms with one selected iMet form and a valid turnover measurement). Next, we also compared abundance measures between the 100 dbTIS/aTIS pairs originating from the same gene. Both analyses indicate that, on average, aTIS were less abundant than dbTIS as referred to in the text (on page 10) and summarised in Table 1 (see also Table 1's legend). More specifically, on page 10 we mention that "Using a Mann-Whitney one sided test we confirmed that on average aTIS variants are less abundant with high significance values using spectral counts (p -value= $4.31e-09$) as well as NSAF scores (p -value= $1.15e-07$). The same was observed when directly comparing gene-specific dbTIS with aTIS proteoforms using a Wilcoxon signed-rank test for paired samples (p -values of $3.44e-09$ and $6.45e-09$ respectively for spectral counts and NSAF, see Table 1)".

To improve clarity, we rephrased the corresponding paragraph in the revised version of the manuscript.

b. Only spectral counts for the two distinct N-terminal peptides were used, and from what I can see in SI Table 1A these were somehow averaged over all time-points? The majority of these counts are quite low <10 , arguing against any useful quantitative data.

As previously reported, using spectral counts as protein abundance estimates is subject to considerable error inherent to MS2 data acquisition, such as different ionization efficiencies between peptides, dynamic exclusion of fragmented peptide ions, reliability of spectral counts from low abundant proteins, reproducibility of chromatographic separation (Bantscheff et al, 2007; Zhou et al, 2012). Despite these drawbacks, spectral counting has been widely used for many relative quantitative proteomics studies. Such data have been gathered in a protein abundance repository called PaxDB, showing that protein abundance estimates can be derived from spectral counting in a reproducible manner, as confirmed by good correlation between protein abundance measured in different cell types and across different species (Wang et al, 2012a; Weiss et al, 2010). From PaxDB we downloaded a shotgun proteomics dataset from Jurkat cells with protein abundances [ppm] derived from spectral counts (data originally published by (Geiger et al, 2012)). We compared these data with our spectral counts obtained for dbTIS proteoforms. Overall, we observed a moderate positive correlation of protein abundances (Pearson correlation coefficient of 0.36), as depicted below in Figure R.1.

□

Figure R.1 | Correlation of protein abundance measured for Jurkat cells using a shotgun (Geiger et al., 2012) and N-terminal COFRADIC approach (this dataset).

We agree with the reviewer that spectral counts <10 might be associated with less reliable quantification (larger error) which was reported previously (Bantscheff et al, 2007; Old et al, 2005). On the other hand, some reports suggest that protein abundances can be determined and compared with high confidence using as little as maximal 4 spectral counts/protein (Old et al, 2005).

Although our dataset is enriched in proteoforms likely expressed at low levels and thus characterised by low spectral counts, we never directly compare individual protein abundances, but rather compare groups. In this case, the uncertainty of measurement is diminished by the number of observations (i.e. 323 aTIS vs. 1,571 dbTIS or 100 aTIS vs. 100 dbTIS). Furthermore, our initial observations of generally lower aTIS spectral counts have now been confirmed at the translational level, as a significantly lower translation initiation signal at the aTIS start codon compared to the dbTIS start codon was detected by ribosome profiling (p -value= 0.011 for the entire population and p -value = 0.00068 for pairs of proteoforms originated from the same gene) (Note: for the use of ribosome profiling we would like to refer to our below comments). Moreover, we observe a good reproducibility of spectral counts obtained for the same proteoforms at different time points (see Figure R.2, Pearson correlation coefficients ranging from 0.55-0.69). The observation that protein quantification can be reproduced in samples that are not directly biological replicates further strengthens our opinion that low spectral counts can be used after careful consideration to draw conclusions regarding proteoform abundances.

Figure R.2] Correlation of \log_2 transformed spectral count values between the 7 time points analysed (time points 0.5 – 48 h are denoted as: count.0.5 – count.48) with their respective Pearson correlation coefficients.

c. In this context, how do the authors know that different N-terminal peptides (proteofroms) have the identical recovery during the COFRADIC enrichment? Have they ever performed any type of recovery experiments using synthetic peptides?

Given the acceptable reproducibility of spectral counts in different experiments presented in Figure R.2, we conclude that the recovery of N-terminal peptides during N-terminal COFRADIC procedure is consistent and reproducible between experiments. By showing a positive correlation of our data to a shotgun proteome analysis (Figure R.1, (Geiger et al, 2012)) we further demonstrate that N-terminal peptides recovered from proteome samples by N-terminal COFRADIC procedure remain representative of their proteoform abundance.

4) The data on protein complexes was interesting and intuitively it makes sense that members of a protein complex have correlated turnover rates. The part of alternative proteoforms and their impact on complex stability was completely unclear and to my mind speculative at best. Unless the authors can show that an alternative proteoform gets incorporated into the same protein complex and as a result affects the turnover rate of the entire complex, which I couldn't find anywhere in the manuscript, then this part of the paper should be toned down or even removed.

In the revised version of the manuscript we now removed the part on the impact of N-terminal proteoforms on protein complexation.

5) While it was nice to see at least a little bit of validation (Figure 7) I found the over-expression studies comparing turnover for dbTIS vs aTIS proteins somewhat crude. To my mind a targeted proteomics assay (MRM) would have been tailor made for these experiments and significantly more accurate.

At least to our knowledge protein turnover has never been studied by MRM and we believe that the study of turnover rates by SILAC pulse labelling fits its purpose much better as compared to the use of MRM, since it solely relies on endogenous protein dynamics and not on spiked in standards. Although MRM enables a targeted quantitative proteomics strategy and is thus ideally for assessing (differences in) the steady state expression levels of N-terminal proteoform pairs, the turnover rates should optimally rely on quantifying the newly synthesized versus the degraded endogenous protein, as was done in this study.

Despite the fact that the classical methods for measuring protein turnover, such as pulse-chase radioactive labeling and inhibition of protein synthesis are not well suited for studying N-terminal proteoforms and their stabilities in unperturbed cellular conditions, we turned to these for validation purposes. This way, and similar as reported in the Schwanhäusser landmark study (Schwanhausser et al, 2011), protein half-lives measured by SILAC and a cycloheximide chase were found to be in good agreement in all cases. Similarly, we first opted to monitor the stabilities of endogenously expressed N-terminal proteoform pairs by means of Western blot analysis following cycloheximide treatment. However, due to the limitation in resolution (at the MW level) when using Western blotting as readout, only a small subset of N-terminal proteoform pairs remained suitable for validation. However of these, typically slow turnover values of both proteoforms were observed precluding a correct delineation of differences in turnover between these relatively long-lived N-terminal proteoforms when making use of cycloheximide. Besides, we encountered difficulties in detecting and unambiguously discriminating between (low abundant) proteoforms (see Figure R.3) at the endogenous level. As such, for validation purposes we ultimately needed to turn to over-expression studies that enabled studying individual N-terminal proteoforms and the comparison of N-terminal proteoform pairs under identical conditions.

Figure R.3 | Endogenous levels of BTF3 proteoforms were monitored in the course of a CHX-chase experiment in Jurkat cells. For the dbTIS proteoform we lacked the 50% turnover information (the dbTIS N-terminal peptide is too short for mass spectrometry based detection). On the other hand, from the two aTIS proteoforms identified in our proteomics study, one was not detected by the antibody (likely the low abundant aTIS (Met 50)).

Minor points

1) Page 5: This reviewer was not completely certain what the authors meant by this sentence "...Important here is that we rely on known N-terminal modifications to unambiguously assign an identified peptide as a proxy for a protein's N-terminus..."?

To discriminate N-termini originating from translation initiation events from those from proteolytic cleavage events, we rely on our previous data regarding the presence (or absence) of in vivo protein modifications obtained during translation; i.e. co-translational modifications. As such, these modifications act as hallmarks to identify a protein N-terminal peptide raised upon translation, here referred to as a proxy for a protein's N-terminus. More specifically and as exemplified in the text, methionine aminopeptidases only remove the initiator Met (iMet) residue if the sidechain of the next amino acid has a small gyration radius like found in alanine (Ala), valine (Val), serine (Ser), threonine (Thr), cysteine (Cys), glycine (Gly) or proline (Pro), meaning that for example asparagine (Asn) and bulky amino acid starting N-termini are not considered here since such events rather point to post-translational proteolytic events. Along the same line, this discarded category of N-termini will typically not carry an in vivo added N-terminal acetyl moiety, a modification occurring on more than 80% of human proteins (Van Damme et al, 2011).

As commented upon by the reviewers and although in the setups analyzed in this study no distinction between in vivo and in vitro Nt-acetylation could be made, it is very important to note that all retained N-termini matched N-termini which are compliant with the rules of N-terminal processing and modification from our previous experiments in which in vivo Nt-acetylation events could be distinguished from in vitro acetylation by the use of stable encoded isotopic variants of NHS-acetate for the blocking of all free protein amine functions in vitro (Van Damme et al, 2014). 1,929 (81.3% of the TIS with MS/MS-based evidence of their modified N-terminal status) of all N-termini reported in this study have previously been identified as being (partially) in vivo Nt-acetylated and this information is now additionally indicated in the supplemental Table EV1. Database annotated N-termini without any available quantitative in vivo Nt-acetylation data were however additionally considered.

Reviewer #2:

MSB 15-6215 Positional proteomics reveals differences in N-terminal proteoform stability Gawron et al.

Synopsis

The authors utilize a double SILAC labeling of Jurkat cells combined with COFRADIC (a technique to isolate N-terminal peptides) to begin to study how proteins with different N-term amino acids, or those produced from different translation start sites, differ in their half lives. They report that the extent of intrinsic disorder was correlated with turnover times.

Minor problems

1. Many experimental details are missing here. For example, it would be helpful for those unfamiliar with the COFRADIC method to at least provide a brief description of how it works, rather than simply referring to previously published protocols.

The methods section has now been modified to include detailed information on the COFRADIC method.

2. The authors do not include sufficient information on how the mass spectrometry was conducted; e.g. what was the resolving power of the orbitrap for MS1? What was the cutoff TIC for taking an MS2? Do each of these analyses represent a single mass spec run? How many mass spec runs were conducted in total? etc. This information must be included to properly judge this work. I would also have really liked to have seen examples of the spectra in the main text, especially for the peptides used for evidence of "alternative" translation initiation sites. On a related note, the use of a single database search engine, and

data presented without any kind of statistical analysis, would be considered somewhat out-of-date by most mass spectrometry professionals (and specialized mass spec journals), at this point in time. A high quality mass spectrometry analysis would now utilize at least two different search engines, followed by a rigorous statistical analysis to assign a confidence value for each peptide and protein identification (e.g. Peptide and Protein Prophet or similar tools).

More detailed information concerning the MS and LC methods applied are now included in the revised methods section. We would also like to point out that all MS/MS spectra have been made accessible in PRIDE with the dataset identifier PXD002091 and 10.6019/PXD002091 and can easily be inspected using for example the PRIDE Inspector tool (Wang et al, 2012b). For this purpose, the following reviewer account may be used: username: reviewer08733@ebi.ac.uk; password: LSrX8ILf. Moreover, the current dataset holds N-termini reported in a previously published dataset of which representative MS/MS spectra of all (alternative) N-termini can be found in the data supplement PDF files of this paper (tables S2A and S2B) at <http://www.mcponline.org/content/13/5/1245/suppl/DC1>.

Concerning the use of multiple search engines we would like to refer the reviewer to our response addressing comment 1.a made by the first reviewer. Further, a strict confidence threshold was being set at 99% confidence and the global false discovery rate (FDR) estimated by searching a decoy database was found to be 1.18% on the spectrum level. Overall, the strategy applied provides a high confidence of the PSMs and protein identifications assigned.

Major concerns

1. The authors indicate on page 7 that the N-terminal peptides reported here were observed in at least 3 time points. Much more detail should be provided here: e.g. I am not convinced that an accurate half-life can be derived when a peptide was detected at times 0.5, 1.5 and 48 hrs. How many such cases are in this dataset, and how did the authors deal with such cases? Is there a statistical argument to be made that this number of observations is sufficient to estimate a proper half-life? Can the authors demonstrate that this is so, using a well-characterized protein as an example? I see this as a significant weakness here, unless the authors can make a convincing argument otherwise.

We monitored several aspects of protein degradation curves to decide if a calculated turnover and half-life is valid. Since 3 data points are the minimal requirement for fitting an exponential model with one parameter and in previous reports on protein stability only 3 time points were analysed (Schwanhausser et al, 2011), we considered proteoforms identified in at least 3 time points. Of note however, in our dataset only 9% of protein turnover rates (177 of 1,941) were calculated based on 3 time points, whereas 50% of all considered values were fully covered in all 7 time points (see Figure R.4).

Next, we evaluated the quality of our estimated model. We excluded modelled half-life values if the curve fitted the experimental data with R^2 coefficient below 0.8 and we excluded cases where no monotonic decrease in the protein M/L ratio was observed (meaning that the protein was not degraded in the measured time frame). Overall, 98% of all protein degradation curves were fitted to our exponential model with R^2 coefficients ≥ 0.8 and the median R^2 value obtained was very high (0.98, see Figure R.5), demonstrating that protein degradation in Jurkat cells follows mono-exponential kinetics.

Although our model explained the bulk of the variability within the data, a certain degree of random variation remained (see Figure R.5). In such cases, a higher coverage of time points might enable a more accurate estimation of protein stability by providing a direct measurement before and after the experimental 50% protein turnover time. In our dataset, the turnover rates calculated for 87% proteoforms are supported by experimental data points that span their measured 50% turnover time, including 56% of the proteoforms found in only 3 time points. Further, there were very few cases when turnover rates were calculated using distant time points (e.g. only 3 cases in the case of the example mentioned by the reviewer i.e. 0.5 h, 1.5 h and 48 h were considered).

Following the reviewer's comment and to further evaluate the relationship between the number of observations and the predictive power of our model, we picked 10 proteoforms identified in 7 time points representing a large variety of turnover rates (Figure R.6A). There are 35 possible combinations of 3 random points when randomly selecting 3 points out of 7. For each peptide, all possible combinations were fitted with an exponential model and used for calculating the 50% turnover time. Next, in line with our original strategy, poor quality models were rejected (R^2 coefficient below 0.8). The remaining valid models were grouped per peptide and represented in Figure R.6B. Clearly, 3 random points are sufficient to reproduce the original 50% turnover time measured for 7 points. The median turnover time based on 3 observations deviated maximally 1.5 h from the turnover calculated using all available data. Importantly, turnover times of less stable proteoforms were very precisely reproduced using 3 randomly selected data points. More stable proteoforms, especially the ones affected by random variability of the data (outliers from the fitted curve, see Figure R.6A) displayed a broader distribution of turnover times (Figure R.6B).

Finally, we compared the stability of database annotated N-terminal proteoforms (dbTIS) measured in our data to a previous study that used the same strategy for turnover time calculation in HeLa cells (whole cell lysates, (Boisvert et al, 2012)). This analysis resulted in a Spearman coefficient ranging from 0.49 for the entire dataset to 0.44 and 0.54 for turnover times measured in 3 and 6-7 time points, respectively. Analogous comparison of dbTIS proteoform half-lives obtained in our study to values reported for the orthologous gene products in mouse (Schwanhausser et al, 2011) revealed a Spearman correlation of 0.48 for the complete dataset, next to 0.59 and 0.47 for half-lives measured using 3 and 6-7 data points, respectively.

To conclude, all these analyses clearly indicate our approach is valid and results in reliable protein turnover estimations, even when stability measurement is based on a limited number experimental data points. Problems inherent to our MS-based strategy, such as the variability of measured SILAC ratios and under-sampling of low abundant proteoforms, were largely overcome by the experimental design (e.g. many time points studied over a broad time frame) and the stringent criteria used to evaluate our model.

Figure R.4 | The number of experimental data points gathered for 1941 N-terminal peptides with successfully calculated turnover times.

Figure R.5 For each N-terminal proteoform quantified in at least 3 time points (1,972), variation of data unexplained by the exponential model was calculated as $(1-R^2)$ and represented in %. The distribution of the unexplained variations is tightly centred around 2% (corresponding to a median $R^2=0.98$) while only a minority of exponential models fitted insufficiently ($R^2<0.8$) and were rejected.

A

B
Figure R.6| Turnover time calculations using a subset of available data points. **A.** We picked 10 proteoforms identified in 7 time points representing a large variety of turnover rates. **B.** For each peptide, all possible combinations of 3 random points were fitted with an exponential model and used for calculating 50% turnover time. Next, models with R^2 coefficient below 0.8 were rejected. The remaining valid models were grouped per peptide and represented as box plots.

On a related note, the authors should at least also discuss the possibility that additional post-translational modifications of the N-terminal peptides (phosphorylation, etc.) could render them undetectable by mass spectrometry: in these cases, the apparent half-life would be short, but not reflect the true half-life of the protein. One additional possibility that should also be discussed here - Do different N-termini contain different numbers of lysine residues that could be ubiquitinated and therefore affect protein half-life?

We agree with the reviewer that various post-translational modifications may have an impact on protein stability. Although these modifications are typically substoichiometric, they further increase N-terminal proteoform diversity in the cell. For this reason, protein turnover measurements are only proxies for the net protein turnover (in this study N-terminal proteoforms), unless all modifications can be monitored. Therefore, it is perhaps more appropriate to refer to the stability of protein pools, as suggested by (Ahmad et al, 2012).

A role for phosphorylation in the context of protein turnover was previously reported (Cambridge et al, 2011). Although turnover rates were not measured directly for phosphopeptides, this study postulated that human proteins with known phosphorylation sites tend to be less stable than non-modified proteins. Another protein turnover study (Ahmad et al, 2012) where phosphorylation events were directly monitored, suggested little or no effect of phosphorylation on the stability of most proteins. Nonetheless, a fraction of modified proteins localized in the nucleolus displayed different stabilities in their phosphorylated form (both increased and decreased stabilities were observed). As such, the role of phosphorylation in determining protein stability is clearly complex and often protein-dependent. Moreover, it is difficult to re-examine our dataset since our sample preparation method did not include phosphatase inhibitors nor did it enrich for phosphopeptides thereby hampering their identification.

Ubiquitination, on the other hand, has a known, direct impact on protein degradation and was previously studied at the proteome-wide level using Jurkat cells as a model system (Stes et al, 2014). Our group reported on a compilation of more than 7,500 *in vivo* ubiquitinated peptides originating from more than 3,300 different proteins (Stes et al, 2014). Using this dataset, and following the reviewer's suggestion, we searched for evidence of ubiquitination possibly occurring within the N-terminal peptides identified in our study. Only peptides corresponding to annotated protein N-termini were used for the comparative analysis. Of the 1,613 dbTIS N-termini with a valid turnover time calculated, 348 were found to contain at least one possible ubiquitination site. Further, we investigated if N-terminal proteoforms holding such ubiquitination sites displayed different stabilities compared to N-terminal proteoforms for which no experimental evidence of ubiquitination was found. As a reference set we selected 388 N-termini indicative of dbTIS proteoforms with valid turnover rates for which no evidence of ubiquitination was found, but containing (a) lysine residue(s). Due to a lack of normal distribution, non-parametrical tests were used (Mann-Whitney test, one-sided) which revealed a significant higher median stability of the ubiquitinated proteoforms compared to the non-ubiquitinated proteoforms ($p=0.0005$). Visualization of the data (see Figure R.7) however shows that this difference is relatively small and that in each case very broad ranges of protein stabilities can be observed.

Figure R.7 Distribution of turnover values for lysine containing ubiquitinated ($N = 348$) and non-ubiquitinated dbTIS indicative N-termini ($N = 388$).

Regarding the higher median stability of ubiquitinated versus non-ubiquitinated proteoforms, several factors may explain such observations. First, since no proteasome inhibitors were used in the ubiquitination study of which note (Stes et al, 2014), ubiquitinated proteins are likely of low abundance and thus only more stably ubiquitinated proteins would reach detectable levels. Moreover, and next to proteasomal degradation, ubiquitin (Ub) tagging of protein substrates may have non-proteolytic consequences, dependent on the number of Ub residues and the topology of Ub-linkages (Kravtsova-Ivantsiv & Ciechanover, 2012). Although proteolysis-promoting Ub-linkages are typically more abundant than other linkage types in human (Dammer et al, 2011), and thus are likely overrepresented in the aforementioned proteomic study (Stes et al, 2014), other types of ubiquitination cannot be ruled out.

To provide more insight, we decided to investigate particular cases of ubiquitinated N-termini. The existence of additional ubiquitination sites may contribute to the difference in turnover observed between longer and shorter proteoforms, as reported in the case of mu opioid receptor (MOP) (Song et al, 2009). An alternative translation initiation site in the MOP mRNA situated upstream (in the 5' leader) gives rise to a longer proteoform with a short half-life due to the additional inclusion of three ubiquitination sites situated within the N-terminal extension. In our dataset 31 dbTIS proteoforms holding unique Ub-sites not

present in a shorter proteoform (aTIS) were found. 11 such proteoforms displayed reduced stability (turnover time decreased by at least 2 hours) compared to their aTIS counterparts (Table R.1). For these proteoforms ubiquitination potentially contributes to the observed differences in stability. We also found 14 proteoform pairs where ubiquitination of the longer variant possibly contributed to an increased stability and 5 cases where non-significant differences in stability between proteoform pairs were observed (<2 h). Do note that because of some large N-terminal truncations, some of the listed N-terminal proteoforms might likely not be isofunctional.

Following the reviewers suggestion we present the ubiquitination analysis of N-terminal proteoforms in the revised version of the manuscript.

Table R.1 | Contribution of ubiquitination to proteoforms stability. Ubiquitination sites previously reported in Jurkat cells (Stes et al, 2014) were mapped onto dbTIS indicative N-termini identified in this study. As such, we found 11 proteoform pairs of which only the database-annotated variant holds (a) previously identified, and thus possible ubiquitination site(s) with lower stability as compared to its N-terminally truncated counterpart.

Sequence with underlined XXGLY – ubiquitination site at Lys (K) (Stes et al, 2014)	Accession	Entry name	Start	Stop	50 % turnover time [h]	Valid turnover measurement	No. of time points	dbTIS/aTIS	Difference in turnover (dbTIS-aTIS) [h]
MDSAGQDINLNSPNK GLL SDSMTDVPVDTGVAAR	O43399	TPD54_HUMAN	1	34	25.08	Yes	7	dbTIS	
TDVPVDTGVAAR			23	34	28.34	Yes	3	aTIS	-3.25
ASNVTN K TDPR	P07910	HNRPC_HUMAN	2	12	30.53	Yes	7	dbTIS	
MIAGQVLDINLAAEPKVN R			74	92	32.69	Yes	7	aTIS	-2.16
MDGIVPDI AVG TKR	P26599	PTBP1_HUMAN	1	14	28.97	Yes	7	dbTIS	
AMAGQSPV L R			176	185	37.43	Yes	7	aTIS	-8.46
PGVTVKDVNQ Q EFVR	P39019	RS19_HUMAN	2	16	31.19	Yes	7	dbTIS	
MVEKDQDG G R			112	121	35.51	Yes	4	aTIS	-4.33
ADIDNKEQSELDQDLDDVVEVEVEEETGEET K LKAR	P55209	NP1L1_HUMAN	2	36	11.69	Yes	4	dbTIS	
MMQNPQILAAL Q ER			42	55	25.43	Yes	6	aTIS	-13.74
MQNPQILAAL Q ER			43	55	25.61	Yes	7	aTIS	-13.92
METE Q PEETFPNTETNGEFG K R	P61978	HNRPK_HUMAN	1	22	21.14	Yes	7	dbTIS	
MEE E QAF K R			27	35	27.57	Yes	7	aTIS	-6.43
AYEPQGGSGYDYSYAG G R			360	377	26.60	Yes	7	aTIS	-5.47
AASAAAAASAAAASAASGSPGPGEGSAGGE K R	Q13263	TIF1B_HUMAN	2	32	23.82	Yes	7	dbTIS	
ALAEGPGAEG P R			580	591	25.96	Yes	7	aTIS	-2.19
MQSNKTFNLE K QNHTPR	Q15233	NONO_HUMAN	1	17	22.48	Yes	7	dbTIS	
MGGAMGIN N R			389	398	25.21	Yes	6	aTIS	-2.73
TKAGSKG G NLR	Q96AG4	LRC59_HUMAN	2	12	19.75	Yes	7	dbTIS	
MKA V QAD Q ER			148	157	33.47	Yes	3	aTIS	-13.72

2. Most translation start sites have been predicted in silico, usually by the presence of a Kozak sequence in a predicted (or experimentally observed) mRNA. This is not even mentioned in the manuscript, but definitely should be for the audience to have a proper understanding of how proteins may end up with different N-termini. The authors suggest that, "...leaky ribosome scanning is likely to be responsible for the majority of downstream translation events observed", but this is not necessarily true - at all. What proportion of the N-terminal peptides identified here could be due to alternative splicing of 5' exons, such that the predicted upstream Met codon is not even present in the resulting mRNA? This would be critical for really understanding what is going on here. How many of the aTIS would be predicted to be the "proper" start site in ESTs (there are huge databases that could be searched for the presence of such examples)? Moreover, given that 2/3 of the observed "aTIS" peptides were detected in the absence of their canonical counterparts, the authors must at least estimate how many of these "alternative" translation start sites might actually be the proper (but mis-annotated) start sites. (i.e. What proportion of these peptides truly represent "alternative" start site usage?) One additional very useful metric to report would be - How many upstream N-terminal peptides have been previously observed in proteomics studies? (This can be easily addressed by searching the many large proteomics databases, freely available.)

In short, while the authors make some possibly interesting observations, they have not done a good job of exploring/explaining why the aTIS may be present in the first place.

First, we would like to apologize for not having stated these points clearly in the original version of the manuscript. In fact, one of our previous studies reporting on an inventory of more than 6,200 TIS-indicative N-termini identified in human and mouse proteomes - holding 1,700 alternative N-termini not annotated in Swiss-Prot - included such an explorative analysis. It consisted of mutagenesis studies and meta-analyses pointing to leaky scanning as the major contributing mechanism leading to TIS selection next to mapping of identified TIS sites with existing Ensembl and/or TrEMBL annotated TIS sites (Van Damme et al, 2014). Importantly, the findings reported in the aforementioned study still hold true for the here reported N-termini as the compilation also included the protein N-termini considered in this study. Mapping data of the N-terminal peptides identified in our proteomics experiment onto the human Swiss-Prot Isoform, UniProt TrEMBL and Ensembl protein databases has now been included in Table EV1C.

Further, while our previous study additionally reported on a comparison with previously published ribosome profiling data on TIS selection referred to as global translation initiation profiling or GTI-seq (Lee et al, 2012), we here opted to perform our own ribosome profiling experiment in Jurkat cells to address this comment even further and thus to study the contribution of splicing and (alternative) translation initiation to N-terminal proteoform expression in more detail.

Ribosome profiling, or Ribo-Seq, enables the study of in vivo protein synthesis by deep sequencing of ribosome-associated mRNA fragments. Ribo-Seq experiments in eukaryotic cells often make use of cycloheximide (CHX) to halt ribosomes during translation. Partial digestion with nucleases is subsequently used to generate the ribosome-protected mRNA sequences that are converted into a sequencing library thereby providing a genome-wide snapshot of actively translated mRNA regions. Additionally, (alternative) translation initiation sites can be detected with sub-codon to single-nucleotide resolution through the use of antibiotics such lactimidomycin (LTM), which exclusively inhibits ribosomes during translation initiation (Lee et al, 2012).

Using ribosome profiling, of the 2,578 N-terminal proteoforms identified by means of N-terminomics, we were able to confirm translation initiation for 89% of dbTIS (1,895 of 2,135) and 29 % of aTIS (130 of 443 aTIS) in Jurkat cells (see Figure R.8A-B). Using a Mann-Whitney one sided test we further confirmed that aTIS detected by Ribo-Seq (130) had significantly lower translation initiation signal ($R_{LTM-CHX}$, see Appendix Supplementary Methods) at the start codon compared to dbTIS (1,895) (p -value= 0.011). The same was observed when directly comparing dbTIS to aTIS variants (31 pairs) detected by Ribo-Seq and originating from the same gene (Wilcoxon signed-rank test for paired samples; p -values of 0.0006782). Further, from the data mapping to the Ensembl and Swiss-Prot Isoforms databases we found that 23% of the aTIS proteoforms (101 out of 443) could likely be explained by a database annotated alternative splice event (see Figure R.8C). Additionally, only 19 aTIS peptides were identified in the knowledgebase TopFIND 3.0 as potential protease cleavage products (Fortelny et al, 2015) (see Figure R.8B). However, of these 7 were disregarded due to Nt-acetylation evidence and/or Ribo-Seq confirmed

TIS selection. This led to a final number of 12 of 443 (3%) aTIS considered as potential proteolytic products (Figure R.8C). The majority of remaining proteoforms (330) is thus likely generated upon leaky scanning or other alternative translation initiation mechanism as previously shown by others and us (Figure R.8C) (Lee et al, 2012) (Van Damme et al, 2014) and see representative examples in Figure R.9A/B/E/F.

Overall, our study demonstrates that N-terminal proteoforms raised upon alternative translation initiation, splicing and alternative iMet processing contribute substantially to the diversity of the human proteome. Indeed, we identified 335 N-terminal proteoforms derived from 133 genes displaying multiple N-termini raised by translation initiation (Figure R.8D). Additionally, we observed 81 proteoforms resulting from partial iMet processing (Figure R.8D). Of note, 63% of all aTIS were identified in the absence of their canonical counterparts.

Figure R.8 | The origin of N-terminal proteoforms. **A-B**. Proteomics-derived translation initiation events were validated at the transcript level using Ribo-Seq data obtained in Jurkat cells and HCT 116 cells (data unpublished). Although in this study, the degree of in vivo Nt-acetylation could not be directly determined, all retained N-termini were compliant with the rules of N-terminal processing from previous experiments (Van Damme et al, 2014). In the case N-termini were found to be in vivo Nt-acetylated (previous MS/MS-based evidence (Van Damme et al, 2014)) this information is indicated. dbTIS and aTIS peptides were further mapped to protein N-termini annotated in Swiss-Prot Isoform, UniProt TrEMBL and Ensembl databases and neo-N-termini (proteolytic products) of the knowledgebase TopFind 3.0. The number of matching N-termini is presented in dark grey. N-terminal peptides supported by at least one source of metadata pointing to translation (Nt-acetylation, Ribo-Seq, Swiss-Prot Isoforms, TrEMBL or Ensembl) were classified as highly confident TIS (indicated in green). **C**. Using experimental and metadata we assigned the most likely origin of aTIS peptides and confirmed that alternative translation initiation within the same transcript (leaky scanning) contributes to the majority of alternative TIS identified. **D**. 2,578 proteoforms were identified including 2,135 dbTIS and 443 aTIS. For the majority of proteins we found a unique translation initiation site (grey). Additionally, 335 N-terminal peptides pointed to proteins with multiple N-termini created by alternative translation initiation (blue), whereas 81 N-termini were generated by alternative iMet processing (green).

Leaky scanning frequently results in initiation at a downstream start codon in the immediate proximity of the first AUG codon (see Figure R.9A) (Van Damme et al, 2014), however, supported by Ribo-Seq data, we found examples of translation initiation by means of leaky scanning at more remote start codons (see Figure R.9B). Relying on the current Ensembl human transcriptome annotation and splice variants present in the Swiss-Prot Isoform database, we were additionally able to confirm the expression of N-terminal proteoforms from alternatively spliced transcripts (see Figure R.9C).

Interestingly, we found 12 aTIS proteoforms being exclusively expressed in the absence of dbTIS, judging from the lack of preceding Ribo-Seq and proteomics signal (Van Damme et al, 2014) (Figure R.9D). The two most likely scenarios leading to the creation of N-terminal proteoforms starting at two consecutive Met residues (e.g. at position 1 and 2), are leaky scanning or alternative iMet processing. Our proteomics data provides evidence of consecutive Met-starting N-termini for 11 genes and matching Ribo-Seq data indicate that translation initiation may occur at both Met residues (Figure R.9E-F).

Figure R.9 Examples of translation initiation events confirmed by Ribo-Seq in Jurkat cells. **A-B.** Alternative translation initiation due to leaky scanning. **C.** Alternative splicing. **D.** The preferential expression of a N-terminally truncated proteoform confirmed by Ribo-Seq data. **E-F.** Evidence for alternative translation initiation occurring at two consecutive AUG codons.

3. Similarly, the authors do a very poor job of relating their findings to previously published papers. e.g. How does the observed half-life of 21.6 hrs compare to previously reported average half-lives of proteins in mammals and model organisms? How do these data on GO categories compare to that

previously observed in mammals and other organisms (e.g. quite a lot of protein turnover work has been conducted on budding yeast, but is not even mentioned here)?

In the original manuscript's discussion section we included a correlation-based comparison of turnover and half-life values obtained from our dataset to previously reported values obtained from human HeLa (Boisvert et al, 2012) and Jurkat cells (Fierro-Monti et al, 2013). Following the reviewer's suggestion, we now performed an extra comparison to mouse data and additionally focussed on the comparison of the enriched GO categories (analysis added to the manuscript). Overall, protein turnover times in Jurkat cells varied from less than 1 to more than 48 hours, with a median turnover time of 21.6 h (Figure 3C). Next to the reported moderate positive correlation, with the Spearman rank correlation coefficient reaching 0.49 for HeLa (Boisvert et al, 2012) and 0.44 for Jurkat data (Fierro-Monti et al, 2013), we found a similar correlation of 0.48 for NIH3T3 mouse data (Schwanhausser et al, 2011) when comparing the turnover values with those of the canonical proteoforms quantified in our study.

Moreover, a median protein half-life of 50.5 hours measured in our study was in good agreement with previously reported values in mouse NIH3T3 cells (47.8 h) (Schwanhausser et al, 2011) and human Jurkat cells (55.8 h) (Fierro-Monti et al, 2013). Further, we observed a highly similar median turnover time (21.6 h) and turnover distribution to estimates obtained in HeLa cells (20 h) (Boisvert et al, 2012).

Based on gene ontology (GO) biological process term enrichment of our data using the Gorilla software (Eden et al, 2009) and considering one proteoform per gene, we found that proteins with common functional annotations displayed comparable cellular stabilities. To reduce the complexity and redundancy of enriched categories we applied REViGIO (Supek et al, 2011) and visualised selected GO terms with FDR q-values lower than 0.05. Unstable proteins were more often involved in mitosis, chromosome segregation, cell cycle and apoptosis next to the regulation of RNA transcription and protein ubiquitination (Figure 4A). On the other hand, RNA splicing, protein translation, folding and transport as well as various metabolic processes of carbohydrates, nucleotides, aldehydes, ketones, glutathione, nitrogen and phosphorus are conducted by more stable proteins (Figure 4B). Along the same line, category enrichment analysis of KEGG pathway terms clearly pointed to increased stability of spliceosome and ribosome components next to RNA degradation, aminoacyl-tRNA biosynthesis, glycolysis and gluconeogenesis, and pentose phosphate pathways (all with FDR values below $9.0E-04$) (Boisvert et al, 2012; Schwanhausser et al, 2011). Pfam data did not point to a significant influence of protein domain composition on protein turnover.

Overall, the results of our annotation enrichment analysis agree very well with previous studies in mouse (Schwanhausser et al, 2011) and human (Boisvert et al, 2012; Fierro-Monti et al, 2013) and thus protein turnover rates seem to be conserved between different cell lines and organisms, especially at the level of protein groups and biological processes.

4. While the introduction provides an overview of the importance of N-terminal acetylation (on Met or other residues, when the initiator Met is removed), it makes no mention at all of the N-end rule. Several decades of research on how the N-terminal residue of a protein influences its half-life (conducted primarily by Varshavsky and colleagues) is not even brought up here.

We agree with the reviewer that the contribution of the N-end rule to protein stability should have been described in detail in the introductory part of the manuscript; i.e. upon first mentioning of the involvement of proteoform N-terminal identity in steering protein stability. As such we now included an introductory paragraph extensively describing the relationship between the in vivo protein half-life and the identity of its N-terminal amino acid(s) (modification) in relationship to the N-end rule.

It is mentioned later on in the manuscript, where the authors go on to argue (with a small, and rather poorly described dataset) that they see no evidence for N-end rule function in their data. In this reviewer's opinion, this could be the most important observation in their manuscript, but is not backed up with a proper, deep analysis of the evidence (i.e. extraordinarily good data would be

required to make this type of extraordinary claim). If their data do not agree with N-end rule predictions, why not?

Although we now removed the somewhat harsh concluding statement that our data does not support the conservation of N-end rule between yeast and human, at least not as a global determinant of protein stability, we gathered statistically convincing data for the role of initiator methionine cleavage in protein turnover, and this irrespective of the rather small sample size. More specifically, a significant impact of iMet processing on turnover rates was confirmed using a Kruskal-Wallis rank sum test ($p < 2.2e-16$) pointing to iMet processing, rather than Nt-acetylation, as a main contributing factor to N-terminal proteoform stability thereby discovering a hitherto unknown property of iMet retention as to increase the stability of N-terminal proteoforms compared to their iMet processed counterparts. This mechanism may exist next to the N-end rule pathways to ensure the fine-tuning of protein stability in eukaryotic cells.

5. Finally, I find the follow-up "verification" to be quite minimal and not very convincing. The authors simply conduct a few Western blotting experiments of endogenous proteins. These blots are not all of high quality; the Westerns have no molecular mass markers; and I would also like to see the entire blot. More importantly, in order to convince this reviewer that the different proteoforms have significant differences in half-life, several examples would have to be engineered for expression in cells (perhaps with C-terminal epitope tags) and proper half-life experiments conducted.

To validate the difference in half-lives of selected proteoform pairs we turned to the use of a more classic method for measuring protein turnover; cycloheximide pulse-chase experiments followed by Western blot analysis. However, since upon incubation of the cells with protein synthesis inhibitors cell toxicity can only be controlled for a relatively short time - and thus proteins with (very) long half-lives are poorly degraded even after prolonged incubation with protein synthesis inhibitors - this precludes a correct delineation/validation of protein half-lives. Therefore, we turned to the validation of relatively short-lived proteins, a category of proteins of typically lower abundance as also demonstrated by the positive correlation between turnover time and protein abundance next to the significantly reduced levels of the top 10% of unstable proteins and higher levels of the 10% most stable proteins in relation to the mean protein abundance (Benjamini-Hochberg FDR of $2.29e-11$ and $7.44e-05$, respectively), and our previous corroborating data demonstrating that the majority of unstable proteins were characterized by lower NSAF values as compared to stable proteins (Koch et al, 2014). For the aforementioned reasons we believe that the lower detection signal of unstable versus stable proteoforms is likely due to the reduced sensitivity of low abundant proteoforms by immunodetection. We now improved the quality of the blots, removed the 48 h cycloheximide treatment conditions, and included molecular weight markers whenever appropriate (see Figure 8). Overall we believe the quality of our Western blot validation results to be well in line with other pioneering protein turnover studies (Schwanhausser et al, 2009) and clearly half-lives measured by SILAC are in good agreement with our Western blotting results despite the inherent differences of the methods used. Finally, expression of the C-terminally tagged N-terminal proteoform pairs of MARE2 and AN32E validated the differential turnover of their dbTIS versus aTIS proteoforms.

Figure 8 | Comparison of protein turnover measured by pSILAC and CHX-chase. **A.** Jurkat cells were treated with 100 µg/ml CHX for 0, 0.5, 1.5, 4, 8, 12 or 24 hours. Protein degradation was monitored by Western blotting and stabilities of several short-lived endogenous proteins were confirmed using antibodies (including lamin B, securin, β-catenin, GCIP interacting protein p29) and compared to stable proteins (such as GAPDH and actin). Proteoform-specific bands were used to calculate protein half-lives which were in a good agreement with turnover times obtained from pSILAC **B-C.** Validation of the differential turnover time of the dbTIS and aTIS derived proteoforms of MARE2 and AN32E. Selective C-terminal V5-tagged proteoforms were overexpressed in HCT116 cells for 24 hours and CHX pulse-chase experiments were performed as described for Jurkat cells. Degradation of overexpressed proteoforms was monitored by an anti-V5 antibody and compared to the turnover of stable proteins such as actin and tubulin.

Reviewer #3:

This is an interesting manuscript that sets out to assess the effect of N-terminal proteoform variation on intracellular stability. The authors use COFRADIC to isolate N-terminal peptides that have been labeled in rather complicated dynamic SILAC experiment.

However, the manuscript actually strays from the implications of the title, and it is really a study of the intracellular stability of proteins in the context of the N-terminus that can be recovered by COFRADIC. Most of the peptides are 'expected' N-termini. It is predominantly therefore an 'actual' N-terminal profile informed turnover analysis.

There are some issues that should be considered. However, with attention and a more critical analysis, the manuscript could become much better, and worthy of publication.

First, the authors use half-life throughout as the measured parameter. This is fundamentally incorrect. The linear parameter is the first order rate constant, and the half-life is a reciprocal term

that distorts the error structure and distribution of the data. The data need to be reexamined in linear parameter space - i.e. the first order rate constant.

We understand the reviewer's point of view. However, we believe that our approach to the interpretation of protein stability measurement using the mathematical inverse of the first order degradation constant (half-life) or turnover time are equally acceptable and valid, as well as widely applied by other researchers (Ahmad et al, 2012; Boisvert et al, 2012; Schwanhausser et al, 2011). Briefly, in our experiments synthesis and degradation curves were fitted to the experimental data (H/L and M/L ratio values, respectively). The crossing point between these curves directly reflects the time needed for the replacement of 50% of protein molecules (i.e. 50% protein turnover time [h]). To account for cell growth, our exponential model of protein degradation incorporated the doubling time measured for Jurkat cells (24 h; also see data presented below). This allowed for the determination of two additional parameters, namely protein degradation rate constants (kdeg) and protein half-life values [h]. A comparison of the 50% protein turnover times to half-life values and kdeg values determined in this study showed that they are perfectly correlated in terms of ranking proteins according to their stability (Spearman rank correlation coefficient of 0.9996 and -0.9996, respectively). The actual values of turnover and half-live time (in hours) are also very similar, although the half-life calculation of stable proteins is more affected by the cell doubling time. Calculated half-lives of the most stable proteins may reach extreme values, while their 50% turnover time is close to the cell doubling time. Importantly, proteins that need more than one cell doubling for the replacements of 50% of their molecules cannot be assigned with a half-life and kdeg value, thereby introducing a bias in any downstream analysis of most stable proteins. Underrepresentation of long-lived proteins when using kdeg values was mentioned previously (Claydon & Beynon, 2012). Once the top 5% of long-lived proteins are disregarded, a very good correlation between 50% turnover time and half-life (Person correlation coefficient of 0.71) can be observed. 50% turnover values can be measured more directly and with greater accuracy, as they do not depend on the cell doubling time determination. More so, 50% turnover time reflects both protein synthesis and degradation. Therefore, we selected 50% turnover time as the desired parameter to perform our subsequent analyses. Importantly, protein turnover values in Jurkat cells are not normally distributed. In consequence, all subsequent analyses were performed using non-parametrical statistics. Non-parametric statistical methods are based on the rank order of protein stability. As we demonstrated above, the assignment of ranks is highly similar for turnover, half-life and kdeg values, which is deemed to result in essentially the same outcome independent of the parameter selected.

To assume a zero order synthesis/first order degradation model (simplistic, but useable) it is important to demonstrate that the loss of isotope follows monoexponential kinetics, and the lack of primary data in the manuscript, or in a supplementary file (why not include an isotope profile, time profile for each protein?).

Using a pulsed SILAC approach, we monitored protein degradation and synthesis based on changes in M/L and H/L isotope ratios (Boisvert et al, 2012). An exponential model with one parameter was fitted to M/L data (degradation curve) using the R software (version 0.98.739) to calculate protein turnover time. All SILAC ratios measured in the experiments are reported in supplemental Table S1A (now renamed to Table EV1A), alongside with parameters reflecting the exponential model fitting: the first order coefficient ("coefB_exp_function" column) and quality of fitness ("R2_exp_function" column), therefore allowing to recreate all degradation profiles. Overall, the loss of medium isotope label followed mono-exponential kinetics, as demonstrated by the fact that 98% of the degradation curves (1,928) successfully fitted such a model, a finding summarised at page 8 of the manuscript: "Using a simple exponential model we calculated turnover times for 1,928 (98%) of the 1,972 proteoforms with minimal R² coefficients ≥ 0.8 ."

To adequately address the reviewer's comment, we additionally document the mono-exponential kinetics assumption in the "Materials and Methods" as well as "Results" sections. Furthermore, representative examples of exponential curve fitting to experimental data of proteoforms with various turnover times (as deduced from their isotope profiles) are shown in Figure R.10A and the degradation profiles of the examples discussed in the manuscript (AIMP2, MARE2 and AN32E, see Figure R.10B) are now presented in the manuscript as a new figure.

Figure R.10 Proteoform degradation profiles. **A.** Representative examples of exponential curve fitting to experimental data of proteoforms with various turnover times. **B.** Degradation profiles of AIMP2, MARE2 and AN32E dbTIS and aTIS proteoforms.

For the N-terminally-informed turnover study, the identity of the actual N-terminus is one piece of information that is readily decoupled from the rate of degradation. Thus, the study could have used unfractionated analyses to recover kdeg - this would have yielded more peptides and more time points, for each protein. The lack of primary data makes it impossible to assess the accuracy with which t05 (really, kdeg) have been determined.

The sole purpose of our experimental setup was to provide, for the first time on a global scale, a link between protein stability and the identity of a protein's N-terminus. As N-terminal proteoforms of the same gene essentially share all peptides except of peptides contained within the N-terminal truncation, we simply can and may not use these shared peptides to obtain proteoform specific

turnover information. On the contrary, our study implies, that in some cases protein turnover rates determined using shotgun proteomics approaches are rather an outcome of the combined stability of different proteoforms. Therefore, we believe that combining pSILAC with an enrichment of N-terminal peptides is the only strategy that allows studying the stability of N-terminal proteoforms and currently cannot be substituted by any other approach. As indicated in our response to a previous comment raised by the reviewer, we provided access to all primary data used in our stability studies, namely SILAC ratios measured in the experiments alongside with parameters reflecting the exponential model fitting; the first order coefficient and quality of fitness.

Secondly, I failed to find a full discussion of the fact that the protein pool used to create the COFRADIC data set is a broken cell preparation. We thus have a mixture of proteins that were destined to end up in different subcellular compartments. Thus, one might posit that the N-end rule only applies to cytosolic proteins.

The methods section has now been modified to include (more) detailed information on the COFRADIC method. The conditions of cell lysis in our experiment indeed lead to organelle disruption, and therefore, our proteome samples may be referred to as total lysates. We agree with the reviewer that the N-end rule might be restricted to certain cellular compartments however, viewing the lack of spatial information, we cannot support this hypothesis with experimental data. Furthermore, we cannot rely on the localization data and gene ontology terms associated with the identified proteins, because many N-terminal proteoforms were reported to reside in different cellular compartments as a consequence of alternative translation initiation (Kazak et al, 2013; Kobayashi et al, 2009; Land & Rouault, 1998; Leissring et al, 2004; Rossmannith, 2011; Suzuki et al, 2010; Van Damme et al, 2014). Due to the lysis conditions applied, the advantage of knowing the N-terminal peptide may be not enough in some cases to precisely determine the stability of a proteoform. The same proteoform residing in different subcellular compartments may have locally different turnover rates. Although we cannot comprehend such level of complexity, viewing the experimental setup used, the Lamond group previously provided evidence for differential stabilities and localisation of proteoform pools (Ahmad et al, 2012; Boisvert et al, 2012). As mentioned in the discussion, studying the differential localisation (and stability) of N-terminal proteoforms is one of our future goals.

Further, N-terminal proteolytic events are common processing events even intracellularly. It is important to clarify whether any of the neo-termini could be due to proteolysis as opposed to alternative splicing. How many of the alternative termini were anticipated based on RNAseq analyses?

As stated in the manuscript text and commented upon in response to previous questions, the discrimination of N-termini originating from translation initiation events from those pointing to proteolytic cleavage events relies on the presence or absence of in vivo protein modifications events occurring during the process of translation (i.e. co-translational modifications). Importantly, non-compliant, proteolytic N-termini (e.g. including mature mitochondrial proteins which lost their N-terminal signal sequence due to proteolysis) were not considered in this study in the first place. Since 1,929 (81.3% of the TIS with MS/MS-based evidence of their modified N-terminal status) of all N-termini reported in this study have previously been identified as being (partially) in vivo Nt-acetylated, only a minority of the here reported N-termini compliant with N-terminal rules (i.e. iMet processing and in vivo Nt-free in the case of for example (M)P- N-termini) could potentially result from proteolysis. Considering the additional metadata obtained (i.e., Ribo-Seq and database mapping) this results in a grand total of 2,566 N-termini (99.5%) indicative of TIS events. To look at the potential contribution of post-translational proteolysis to the generation of the remaining alternative N-termini identified, we performed a comparison with proteolytic cleavages contained within TopFIND 3.0, a public knowledgebase and analysis resource for protein termini and protease processing which holds more than 8,000 cleavages in human proteins (Fortelny et al, 2015). For 7 of the 19 cleavage sites matching N-termini identified in this study, metadata pointed to translation initiation, overall leaving only 12 N-termini matching previously identified cleavages. Finally, to study the contribution of splicing and (alternative) translation initiation to N-terminal proteoform expression, we would like to refer to our previous answer raised in response to comment 2 made by

reviewer 2, where we show that 71% of all aTIS are supported by additional metadata pointing to their translational origin.

The obligatory N terminal acetylation does of course converge the acetylated and non-acetylated proteins into one, consistently behaving pool. However, it would have been helpful to repeat the analysis with ¹³C-acetic anhydride, to be able to discriminate the natural from the chemical modifications. These data could then be combined with the turnover data, knowing which was naturally acetylated.

We addressed this issue in response to previous comments. In the setups analyzed in this study no distinction between in vivo and in vitro Nt-acetylation could be made. However, it is very important to note that all retained N-termini matched N-termini compliant with the rules of N-terminal processing from previous experiments in which in vivo Nt-acetylation events could readily be distinguished (and quantified) from in vitro acetylation by the use of stable encoded isotopic variants of NHS-acetate for the blocking of all free protein amine functions in vitro (Van Damme et al, 2014). In the case N-termini were previously found to be in vivo Nt-acetylated, this information is now additionally indicated in an extra column in Table EV1A entitled 'in vivo Nt-Ac (MS/MS)'.

A smaller part of the data set refer to alternative N-terminal proteoforms (assumed to be due to alternative splicing, but other routes to neo-termini?) This is a very interesting part of the work, but it is a minor part of the work. The turnover study alone is not a hugely original study, even if driven by a single peptide/n-terminal approach. The proteoform segment would require much more analysis, and ideally, a stronger indication of the quality of the turnover rates. For this subset, accurate measure of kdeg, with multiple peptides, should be quite feasible.

We disagree with the reviewer, since based on our data including comparative analyses of 100 N-terminal proteoform pairs - the majority raised upon (alternative) translation initiation - provides unprecedented insight into how (alternative) translation can regulate protein stability. Currently, we are unaware of any other strategy/report enabling such detailed insight. Besides, all other remarks were addressed in previous comments or in our previously published work (Van Damme et al, 2014).

Technical points.

It is important to be sure that the labeling protocols work (100% label, complete transition to unlabelling etc.). Some supplementary data affirming this would be helpful. Some indication of the quality of the COFARDIC protocol is needed - how many internal peptides were present in the sample?

As suggested by the reviewer, we performed a validation experiment of the labelling protocol. Jurkat cells cultured in regular RPMI medium were transferred to a SILAC RPMI medium containing Arg⁶ L-arginine and cultured for 7 days until complete label incorporation (i.e. cells were cultured for at least six population doublings to ensure complete incorporation of the labelled arginine; see also cell doubling calculation data presented below). Subsequently, cells were transferred to a SILAC RPMI medium containing Arg⁰ L-arginine for 7 next days, to achieve complete unlabelling. The cell culture was sampled at different time points during this process, isolated proteomes digested by trypsin and analysed on an Orbitrap Velos mass spectrometer. The isotope replacement was monitored by quantifying the identified MS-spectra using Mascot Distiller. As expected, these data confirm that the conditions of culture used are appropriate for the complete replacement of isotopically labelled arginine in Jurkat cells and that the labelling efficiency clearly follows the cell doubling (i.e. after 3 days of unlabelling, on average 11.9% of the heavy label remained which is expected from the expected cell doubling time of 24 h calculated (i.e. 1/8)). The distribution of SILAC ratios at the different stages of labelling is presented in a new supplemental figure (see also Figure R.11 below).

Figure R.11 | SILAC labelling of Jurkat cells.

I think the doubling time of the Jurkat cells should have been cited. If it is 20h approx. (typical) then this means that most proteins are actually replaced by dilution, not by degradation. Figure 3c should have the doubling time (in h) marked clearly on the graph.

As requested by the reviewer, we present the results of doubling time determination in Jurkat cells cultivated in SILAC RPMI medium (see Figure R.12 below). The doubling time in these conditions was found to be 24 hours and this value was used for the calculation of protein degradation constants and their corresponding half-lives. We included this information in the revised version of the manuscript.

Figure R.12 | Experimental assignment of Jurkat cell doubling time. Density of cell culture was monitored in triplicate over the course of 48 hours and fitted with an exponential model using the Doubling Time Online Calculator (<http://www.doubling-time.com/compute.php>). Cell doubling time

was calculated from the assigned growth rate (0.0288) as follows: $\ln(2)/0.0288$ and represented in hours.

I have some nervousness about the contention that incubating a cell in cycloheximide for 48h would result in a living cell that maintained the machinery and energetics for sustained degradation. Perhaps the authors would address this. They are not dying at 48h, they are dead!

We agree with the reviewer, that upon incubation of the cells with protein synthesis inhibitors, cell toxicity can only be controlled for a relatively short time. This precludes a precise delineation/validation of long protein half-lives (see β -actin and GAPDH in Figure 8A), therefore we turned to the validation of relatively short-lived proteins (Figure 8A, securing, SYF2, β -catenin, lamin B). As indicated in our answer to comment 5 made by reviewer 2, we now improved the quality of the blots and removed the 48 h CHX treatment conditions. Overall, we believe the quality of our Western blot validation results to be well in line with blots presented in other protein turnover studies (Schwanhausser et al, 2009) and clearly half-lives measured by SILAC are in good agreement with our Western blotting results despite the inherent differences of the methods used.

By definition, all peptides are 'one hit wonders'. It is thus critical that we be given a measure of peptide quality. Other journals would request the MS/MS spectrum of each of these peptides.

As commented upon previously (comment 2, reviewer 1), we would like to point out that all MS/MS spectra have been made available through the PRIDE database and can easily be inspected using for example the PRIDE Inspector tool (Wang et al, 2012b) with the dataset identifier PXD002091 and 10.6019/PXD002091. The following reviewer account may be used: username: reviewer08733@ebi.ac.uk; password: LSrX8ILf. Additionally, peptides presented in the current study were characterized in our previous work that reported a compilation of human and mouse dbTIS and aTIS, including all highest scoring MS/MS spectra in a pdf format (Van Damme et al, 2014).

The authors should have included a text file or excel sheet of protein, peptide, peptide quality/score, and kdeg.

The information on peptide and protein identifications as well as kdeg values requested by the reviewer is available in the Supplemental Table EV1A. As suggested by the reviewer, we have now added Mascot scores of the highest scoring MS/MS-spectra per peptide.

References:

Ahmad Y, Boisvert FM, Lundberg E, Uhlen M, Lamond AI (2012) Systematic analysis of protein pools, isoforms, and modifications affecting turnover and subcellular localization. *Mol Cell Proteomics* **11**: M111 013680

Bantscheff M, Schirle M, Sweetman G, Rick J, Kuster B (2007) Quantitative mass spectrometry in proteomics: a critical review. *Anal Bioanal Chem* **389**: 1017-1031

Boisvert FM, Ahmad Y, Gierlinski M, Charriere F, Lamont D, Scott M, Barton G, Lamond AI (2012) A quantitative spatial proteomics analysis of proteome turnover in human cells. *Mol Cell Proteomics* **11**: M111 011429

Cambridge SB, Gnad F, Nguyen C, Bermejo JL, Kruger M, Mann M (2011) Systems-wide proteomic analysis in mammalian cells reveals conserved, functional protein turnover. *J Proteome Res* **10**: 5275-5284

Claydon AJ, Beynon R (2012) Proteome dynamics: revisiting turnover with a global perspective. *Mol Cell Proteomics* **11**: 1551-1565

- Colaert N, Van Huele C, Degroeve S, Staes A, Vandekerckhove J, Gevaert K, Martens L (2011) Combining quantitative proteomics data processing workflows for greater sensitivity. *Nat Methods* **8**: 481-483
- Dammer EB, Na CH, Xu P, Seyfried NT, Duong DM, Cheng D, Gearing M, Rees H, Lah JJ, Levey AI, Rush J, Peng J (2011) Polyubiquitin linkage profiles in three models of proteolytic stress suggest the etiology of Alzheimer disease. *J Biol Chem* **286**: 10457-10465
- Eden E, Navon R, Steinfeld I, Lipson D, Yakhini Z (2009) GOrilla: a tool for discovery and visualization of enriched GO terms in ranked gene lists. *BMC Bioinformatics* **10**: 48
- Fierro-Monti I, Racle J, Hernandez C, Waridel P, Hatzimanikatis V, Quadroni M (2013) A novel pulse-chase SILAC strategy measures changes in protein decay and synthesis rates induced by perturbation of proteostasis with an Hsp90 inhibitor. *PLoS One* **8**: e80423
- Fortelny N, Yang S, Pavlidis P, Lange PF, Overall CM (2015) Proteome TopFIND 3.0 with TopFINDER and PathFINDER: database and analysis tools for the association of protein termini to pre- and post-translational events. *Nucleic Acids Res* **43**: D290-297
- Geiger T, Wehner A, Schaab C, Cox J, Mann M (2012) Comparative proteomic analysis of eleven common cell lines reveals ubiquitous but varying expression of most proteins. *Mol Cell Proteomics* **11**: M111 014050
- Kazak L, Reyes A, Duncan AL, Rorbach J, Wood SR, Brea-Calvo G, Gammage PA, Robinson AJ, Minczuk M, Holt IJ (2013) Alternative translation initiation augments the human mitochondrial proteome. *Nucleic Acids Res* **41**: 2354-2369
- Kobayashi R, Patenia R, Ashizawa S, Vykoukal J (2009) Targeted mass spectrometric analysis of N-terminally truncated isoforms generated via alternative translation initiation. *FEBS Lett* **583**: 2441-2445
- Koch A, Gawron D, Steyaert S, Ndah E, Crappe J, Keulenaer SD, Meester ED, Ma M, Shen B, Gevaert K, Criekinge WV, Damme PV, Menschaert G (2014) A proteogenomics approach integrating proteomics and ribosome profiling increases the efficiency of protein identification and enables the discovery of alternative translation start sites. *Proteomics*
- Kravtsova-Ivantsiv Y, Ciechanover A (2012) Non-canonical ubiquitin-based signals for proteasomal degradation. *J Cell Sci* **125**: 539-548
- Land T, Rouault TA (1998) Targeting of a human iron-sulfur cluster assembly enzyme, nifs, to different subcellular compartments is regulated through alternative AUG utilization. *Mol Cell* **2**: 807-815
- Lee S, Liu B, Lee S, Huang SX, Shen B, Qian SB (2012) Global mapping of translation initiation sites in mammalian cells at single-nucleotide resolution. *Proc Natl Acad Sci U S A* **109**: E2424-2432
- Leissring MA, Farris W, Wu X, Christodoulou DC, Haigis MC, Guarente L, Selkoe DJ (2004) Alternative translation initiation generates a novel isoform of insulin-degrading enzyme targeted to mitochondria. *Biochem J* **383**: 439-446

- Old WM, Meyer-Arendt K, Aveline-Wolf L, Pierce KG, Mendoza A, Sevinsky JR, Resing KA, Ahn NG (2005) Comparison of label-free methods for quantifying human proteins by shotgun proteomics. *Mol Cell Proteomics* **4**: 1487-1502
- Paoletti AC, Parmely TJ, Tomomori-Sato C, Sato S, Zhu D, Conaway RC, Conaway JW, Florens L, Washburn MP (2006) Quantitative proteomic analysis of distinct mammalian Mediator complexes using normalized spectral abundance factors. *Proc Natl Acad Sci U S A* **103**: 18928-18933
- Rossmannith W (2011) Localization of human RNase Z isoforms: dual nuclear/mitochondrial targeting of the ELAC2 gene product by alternative translation initiation. *PLoS One* **6**: e19152
- Schwanhaussner B, Busse D, Li N, Dittmar G, Schuchhardt J, Wolf J, Chen W, Selbach M (2011) Global quantification of mammalian gene expression control. *Nature* **473**: 337-342
- Schwanhaussner B, Gossen M, Dittmar G, Selbach M (2009) Global analysis of cellular protein translation by pulsed SILAC. *Proteomics* **9**: 205-209
- Song KY, Choi HS, Hwang CK, Kim CS, Law PY, Wei LN, Loh HH (2009) Differential use of an in-frame translation initiation codon regulates human mu opioid receptor (OPRM1). *Cell Mol Life Sci* **66**: 2933-2942
- Stes E, Laga M, Walton A, Samyn N, Timmerman E, De Smet I, Goormachtig S, Gevaert K (2014) A COFRADIC protocol to study protein ubiquitination. *J Proteome Res* **13**: 3107-3113
- Supek F, Bosnjak M, Skunca N, Smuc T (2011) REVIGO summarizes and visualizes long lists of gene ontology terms. *PLoS One* **6**: e21800
- Suzuki Y, Holmes JB, Cerritelli SM, Sakhuja K, Minczuk M, Holt IJ, Crouch RJ (2010) An upstream open reading frame and the context of the two AUG codons affect the abundance of mitochondrial and nuclear RNase H1. *Mol Cell Biol* **30**: 5123-5134
- Van Damme P, Gawron D, Van Crielinge W, Menschaert G (2014) N-terminal proteomics and ribosome profiling provide a comprehensive view of the alternative translation initiation landscape in mice and men. *Mol Cell Proteomics* **13**: 1245-1261
- Van Damme P, Hole K, Pimenta-Marques A, Helsens K, Vandekerckhove J, Martinho RG, Gevaert K, Arnesen T (2011) NatF contributes to an evolutionary shift in protein N-terminal acetylation and is important for normal chromosome segregation. *PLoS Genet* **7**: e1002169
- Wang M, Weiss M, Simonovic M, Haertinger G, Schrimpf SP, Hengartner MO, von Mering C (2012a) PaxDb, a database of protein abundance averages across all three domains of life. *Mol Cell Proteomics* **11**: 492-500
- Wang R, Fabregat A, Rios D, Ovelleiro D, Foster JM, Cote RG, Griss J, Csordas A, Perez-Riverol Y, Reisinger F, Hermjakob H, Martens L, Vizcaino JA (2012b) PRIDE Inspector: a tool to visualize and validate MS proteomics data. *Nature biotechnology* **30**: 135-137
- Weiss M, Schrimpf S, Hengartner MO, Lercher MJ, von Mering C (2010) Shotgun proteomics data from multiple organisms reveals remarkable quantitative conservation of the eukaryotic core proteome. *Proteomics* **10**: 1297-1306

Yu W, Taylor JA, Davis MT, Bonilla LE, Lee KA, Auger PL, Farnsworth CC, Welcher AA, Patterson SD (2010) Maximizing the sensitivity and reliability of peptide identification in large-scale proteomic experiments by harnessing multiple search engines. *Proteomics* **10**: 1172-1189

Zhou W, Liotta LA, Petricoin EF (2012) The spectra count label-free quantitation in cancer proteomics. *Cancer Genomics Proteomics* **9**: 135-142

2nd Editorial Decision

22 December 2015

Thank you again for submitting your work to Molecular Systems Biology. We apologize for the lengthy process. We have now finally heard back from reviewer #1. While reviewer #2 should have returned a report as well, we prefer to make a decision now rather than delaying further the process. We are now globally satisfied with the modification made as described in your reply letter and I am pleased to inform you that we will be able to accept your study for publication pending the following minor points:

- Please include a "Data availability" section at the end of Materials and Methods, that details the accession numbers of the deposited MS proteomics data as well as the Ribo-Seq data.

- We would suggest to include the whole Materials & Methods section in the main paper (it is not included in the character count anyway) rather than in the Appendix.

- As you may have noticed, we recently replaced Supplementary Information by Expanded View (EV, see examples in <http://msb.embopress.org/content/11/6/812>). In this format, a limited number of Supplementary Figures (max 5) can be integrated in the article as so-called "EV figures" that are interactively collapsible/expandable and will be typeset by the publisher. In this case, the figures should be cited as 'Figure EV1, Figure EV2' etc... in the text and their respective legends should be added to the main text after the legends of regular figures. The illustrations should be provided as separate files. In the case of this study, since you have only 5 Figures in the Appendix, we would suggest to remove them from the Appendix and relabel them as Figure EV1, EV2, etc.. in the text and add the legend accordingly in the Word file.

- For figure panels that show Western blot results (Fig 7), we would also kindly ask you to include the respective source data in the form of the original uncropped images. There is no need to assemble the individual images in a figure--the files can simply be labeled according to the protein assayed and the corresponding panel in the figure (eg "figure7-panelA-beta-actin.jpg"). All the files associated to a given panel can be zipped together and submitted as "Source data for Figure 7A", "Source data for Figure 7B" etc.. using 'dataset' as object type when uploading the files. (Additional information on source data is available in the "Guide for Authors" section at <http://msb.embopress.org/authorguide#sourcedata>).

Revision - authors' response

07 January 2016

We would like to thank you for accepting our manuscript **MSB-15-6662** entitled 'Positional proteomics reveals differences in N-terminal proteoform stability' by Daria Gawron and colleagues for publication in *Molecular Systems Biology*. We would also like to thank the reviewer for expressing interest in our study. In response to the remaining comments, the revised version of our manuscript comprises the following data and action points taken:

- We included the entire “Materials and Methods” section in the main article.
- We included a “Data availability” section at the end of the “Materials and Methods” section.
- We placed an author contributions statement after the “Acknowledgements” section.
- We moved all figures from Supplementary Information to Expanded View (Figure EV1 – EV5). We cited them appropriately in the main text. Their respective legends were added to the main text after the legends of regular figures. Illustrations were provided as separate files.
- For figure panels that show Western blot results (Fig 8), we included their respective source data in the form of the original uncropped images labeled according to the protein assayed with annotation of protein mass standards and well content. Files associated to a given panel were zipped together for submission.
- For the synopsis section we provided a short text summarizing the study, three 'bullet points' highlighting the main findings and a 'thumbnail image' to serve as a visual summary of the article. We hope that our revised manuscript can be found acceptable for publication in *Molecular Systems Biology* and we are looking forward to your reply.